# Arabidopsis guard cell chloroplasts import cytosolic ATP for starch turnover and stomatal opening

Shey-Li Lim [1,5], Sabrina Flütsch [2,5], Jinhong Liu [1], Luca Distefano [2], Diana Santelia [2✉] & Boon Leong Lim [1,3,4✉]

Stomatal opening requires the provision of energy in the form of ATP for proton pumping across the guard cell (GC) plasma membrane and for associated metabolic rearrangements. The source of ATP for GCs is a matter of ongoing debate that is mainly fuelled by controversies around the ability of GC chloroplasts (GCCs) to perform photosynthesis. By imaging compartment-specific fluorescent ATP and NADPH sensor proteins in Arabidopsis, we show that GC photosynthesis is limited and mitochondria are the main source of ATP. Unlike mature mesophyll cell (MC) chloroplasts, which are impermeable to cytosolic ATP, GCCs import cytosolic ATP through NUCLEOTIDE TRANSPORTER (NTT) proteins. GCs from *ntt* mutants exhibit impaired abilities for starch biosynthesis and stomatal opening. Our work shows that GCs obtain ATP and carbohydrates via different routes from MCs, likely to compensate for the lower chlorophyll contents and limited photosynthesis of GCCs.

[1] School of Biological Sciences, University of Hong Kong, Hong Kong, China. [2] Institute of Integrative Biology, ETH Zürich, Zürich, Switzerland. [3] HKU Shenzhen Institute of Research and Innovation, Shenzhen, China. [4] State Key Laboratory of Agrobiotechnology, The Chinese University of Hong Kong, Hong Kong, China. [5] These authors contributed equally: Shey-Li Lim, Sabrina Flütsch. ✉email: dsantelia@ethz.ch; bllim@hku.hk

Stomata are tiny pores on the leaf surface, each surrounded by a pair of guard cells (GCs). In higher plants, stomatal opening allows the inflow of carbon dioxide ($CO_2$) and the diffusion of oxygen and water, thereby driving photosynthesis and the transpiration stream from the roots to the leaves[1,2]. Stomata are closed when GCs are flaccid and open when the GCs become turgid through the influx of inorganic and organic ions, such as potassium ($K^+$), chloride ($Cl^-$), nitrate ($NO_3^-$), and malate ($Mal^{2-}$)[3].

Since GCs lack plasmodesmata, the flux of ions in and out of GCs must be mediated by transporters or ion channels at the plasma membrane (PM)[4,5]. At dawn, the blue light (BL)-activated PM proton ($H^+$) pump $H^+$-ATPase transports $H^+$ from the cytosol to the apoplast, at the expense of hydrolysing ATP, causing hyperpolarisation of the PM and activation of voltage-gated $K^+$ channels[6]. The resulting influx of $K^+$ ions lowers GC water potential, promoting osmotic water flow, which increases turgor pressure of GCs and opens stomata[7]. Uptake by GCs of mesophyll-derived sugars via PM sugar/$H^+$ cotransporters also contributes to efficient stomatal opening at dawn, although the precise function of sugars within GCs is not fully understood[8–11]. In parallel to the activation of membrane ion transport, BL triggers rapid starch degradation in GC chloroplasts (GCCs), which contributes to fast stomatal opening kinetics[12,13].

Given that PM hyperpolarization and sugar transport largely rely on $H^+$-ATPase activity, stomatal opening is energetically costly, consuming large amounts of cytosolic ATP[4,14,15]. Despite the central role of ATP in energizing stomatal movements, the source of ATP for GCs has been a matter of debate for several decades. Theoretically, ATP can be provided by glycolysis, mitochondrial respiration, and photosynthesis. GCs possess strong glycolytic activity in the light[16–18] and *Arabidopsis thaliana* (Arabidopsis) mutants with defective glycolysis have impaired BL-induced stomatal opening[19]. GC protoplasts (GCPs) also have higher respiratory rates in the dark compared to mesophyll cell protoplasts (MCPs)[20,21], and exogenous application of respiratory inhibitors, such as sodium azide or oligomycin, to *Commelina benghalensis* epidermal strips reduces light-induced stomatal opening[22]. These findings suggest that oxidative phosphorylation is a basic source of energy in GCs and are in line with the observation that GCs contain unusually large numbers of mitochondria[23].

On the other hand, GCs have fewer and smaller chloroplasts with lower chlorophyll contents and less granal stacking compared to MCs[24], calling into question the ability of GCCs to perform photosynthesis. Several studies reported that GCCs cannot fix $CO_2$, mostly due to the low content and activity of ribulose-1,5-biphosphate carboxylase/oxygenase (RubisCO)[25–28]. However, other studies detected Calvin–Benson–Bassham (CBB) cycle activity in GCs, demonstrating $CO_2$ uptake into 3-phosphoglycerate (3-PGA) and ribulose 1,5-bisphosphate[29–31]. High-resolution chlorophyll fluorescence imaging further revealed that the GCC electron transport chain (cETC) is functional and RubisCO is a major sink for the end products of electron transport[32]. These findings have been confirmed recently by mass spectrometry-based $^{13}C$-isotope labelling experiments in tobacco (*Nicotiana tabacum*) epidermal fragments, indicating that GCCs are able to fix $CO_2$ by both RubisCO and the anaplerotic reactions catalyzed by phospho*enol*pyruvate carboxylase (PEPc)[17].

It is clear from these studies that GC photosynthesis is a highly controversial topic. Although there seems to be agreement that it takes place, it is unclear to what extent GCCs contribute to the pool of GC sugars or ATP production for $H^+$ extrusion during stomatal opening[33–35]. Some reports suggested that GCCs supply ATP to the cytosol, which is used for $H^+$ pumping[14,15,36].

Sophisticated fluorescence analyses combined with patch-clamp experiments demonstrated that GC photosynthesis and PM $K^+$ channel activity in single *Vicia faba* GCPs depend on cytosolic ATP[37].

Here, we combined photosynthetic and respiratory inhibitors with fluorescent protein sensing to investigate the ability of Arabidopsis GCs to perform photosynthesis. Following subcellular changes in fluorescent ratios of cytosol- and plastid-targeted versions of ATP[38], NADPH[39], and pH[40] sensors in response to illumination, we show that GCs produce negligible amounts of ATP and NADPH through photosynthesis. The majority of ATP is provided by oxidative phosphorylation in mitochondria, which is activated in the light. Additional physiological and biochemical experiments further demonstrated that GC metabolism is highly adapted to supply the demand of energy for stomatal movements. Our results provide several clues towards solving the longstanding debate around the specialized GC metabolism and move research in this field forward.

## Results

**Illumination stimulates detectable ATP and NADPH production in MCCs but not in GCCs**. To determine the extent to which GCCs produce energy in the form of ATP and NADPH in response to light, we examined the dynamic changes of stromal ATP and NADPH levels in Arabidopsis GCCs at three different time points (end of night (EoN) and 2 h and 8 h into the day). Mature leaves were collected from 3-week-old wild-type (WT) plants expressing plastid (TKTP)-targeted versions of a $MgATP^{2-}$-specific Förster resonance energy transfer (FRET)-based sensor (AT1.03)[38], a NADPH sensor (iNAP4)[39] or a pH sensor (cpYFP)[40]. Leaves were illuminated for 3 min under a confocal microscope, and changes in ratios upon dark-to-light transition were determined through ratiometric image analysis. Illumination stimulated ATP and NADPH production and alkalinization in mesophyll cell chloroplasts (MCCs) but not in GCCs, independently of the time of day (Fig. 1, Supplementary Fig. 1). We further recorded changes in AT1.03 FRET ratios upon illumination in rosette leaves at two different developmental stages with comparable results (Supplementary Fig. 2). Considering that in our experiments changes in fluorescence emission ratios were within each sensor's optimal range of detection (Fig. 1), these data suggest that, compared to MCCs, photosynthetic production of ATP and NADPH in Arabidopsis GCCs is limited.

**GCCs have higher levels of NADPH in the dark than MCCs**. To expand our analysis, we compared real-time dynamic changes of sensor ratios in the chloroplast stroma of MCs, GCs, and root cells. The stromal $MgATP^{2-}$, NADPH and pH sensors in GCs and root cells were not responsive to light (Fig. 2), in line with the idea that GCs have several features of sink cells. However, GCCs contained larger amounts of ATP and NADPH and had a more alkaline pH compared to root plastids (Fig. 2b, d, f). The levels of NADPH at EoN in GCCs were even higher than those of MCCs, although no noticeable increases in stromal NADPH were observed in GCCs upon illumination (Fig. 1c, d, Fig. 2d). To confirm that the NADPH sensor was active in GCCs, we infiltrated plants expressing the TKTP-iNAP4 sensor with the oxidizing agents hydrogen peroxide ($H_2O_2$) or menadione as a control experiment. Both $H_2O_2$ and menadione suppressed the stromal NADPH sensor ratio in GCs and MCs, further indicating that chloroplasts in both cell types contain substantial amounts of NADPH in the dark (Fig. 3a, b).

To understand how GCCs produce NADPH, we treated dark-adapted plants expressing TKTP-iNAP4 with 6-aminonicotinamide

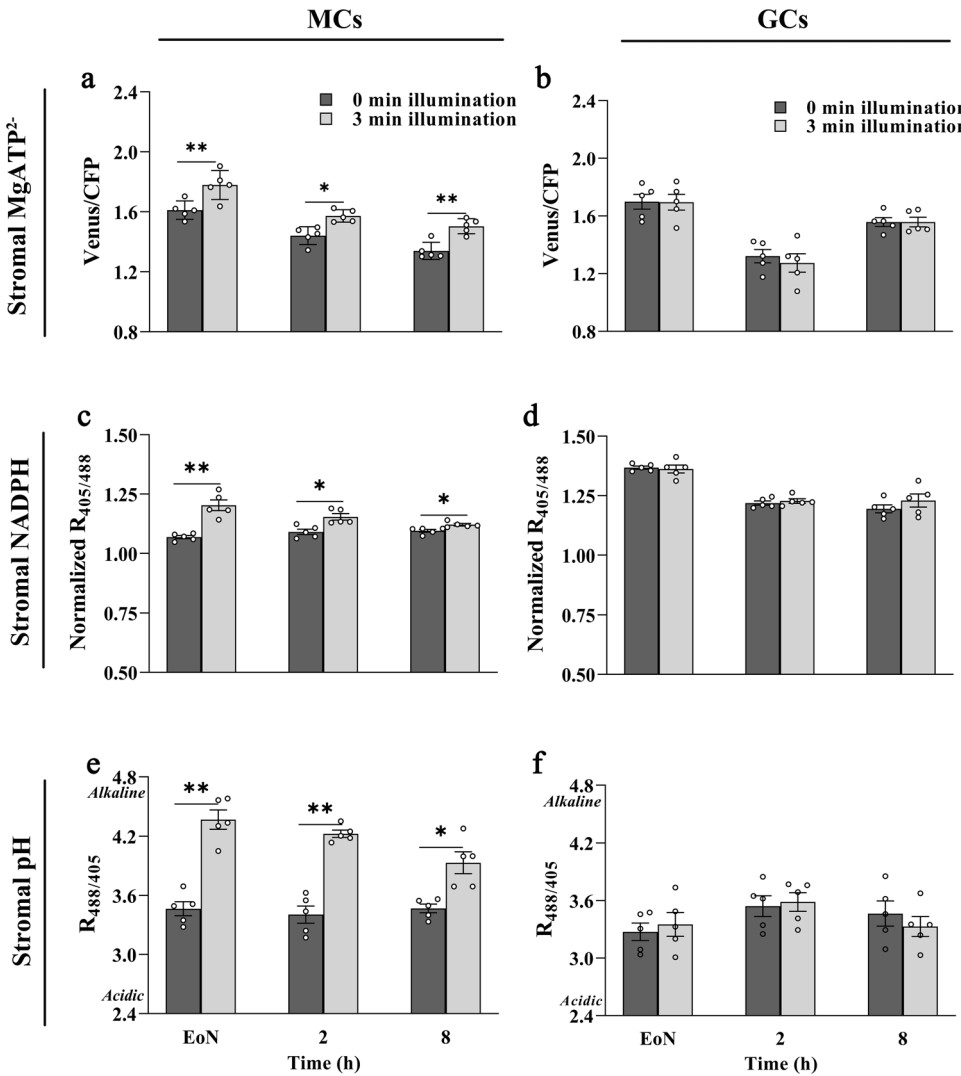

**Fig. 1 Illumination induces detectable ATP and NADPH biosynthesis in mesophyll cell chloroplasts but not in guard cell chloroplasts.** The 3rd and 4th leaves of 20- to 22-day-old wild-type Arabidopsis plants expressing TKTP-AT1.03 (ATP sensor), TKTP-iNAP4 (NADPH sensor) or TKTP-cpYFP (pH sensor) were collected at three different time points (EoN, 2 h and 8 h into the day). **a, b** Stromal AT1.03 signals (p-values of panel a: EoN = 0.012, 2 h = 0.036, and 8 h = 0.010), **c, d** Stromal iNAP4 signals (p-values of panel c: EoN = 0.004, 2 h = 0.032, and 8 h = 0.015), and **e, f** Stromal cpYFP signals (p-values of panel e: EoN = 0.002, 2 h = 0.002, and 8 h = 0.017) from mesophyll cells (MCs, **a**, **c**, **e**) and guard cells (GCs, **b**, **d**, **f**) in response to white light illumination at 216 µmol m$^{-2}$ s$^{-1}$ for 180 s. The iNAP results presented here were normalized to TKTP-iNAPc. EoN, End of night. Asterisks indicate significant statistical differences (*$P < 0.05$, **$P < 0.01$) before and after 180 s of illumination, as determined by a paired $t$-test, two-tailed ($n = 5$; mean ± SEM). Source data are provided as a source data file.

(6-AN). 6-AN is an inhibitor of 6-phosphogluconate dehydrogenase used to block the NADPH-producing oxidative pentose phosphate pathway (OPPP), normally operating in plastids in the dark[41]. Treatment with 6-AN reduced stromal NADPH level in both GCs and MCs (Fig. 3c, d), suggesting that the OPPP pathway is a major source of NADPH in the dark, particularly in GCCs, in which we showed cETC activity is minimal and NADPH levels are elevated at EoN.

**Respiratory inhibitors deplete GC cytosolic ATP levels.** Our finding that GCCs have limited cETC capacity, along with the observation that the ratio of chloroplasts to mitochondria is much lower in GCs than in MCs[23], led us to reason that mitochondria may be the main suppliers of ATP in GCs. Our transgenic lines expressing a cytosol-targeted version of the AT1.03 sensor offered the possibility to directly test the contribution of mitochondrial ATP production to cytosolic ATP levels. We tracked changes in

cytosolic AT1.03 FRET ratios in GCs and MCs of mature leaves after 1 h dark incubation in the presence or absence of several inhibitors of the mitochondrial electron transport chain (mETC). Inhibition of either complex I (rotenone) or complex II (thenoyltrifluoroacetone, TTFA), which feed electrons into the mETC by oxidizing NADH and succinate, respectively, only partially reduced cytosolic MgATP$^{2-}$ concentrations and had a similar effect on MCs and GCs (Fig. 3e, f). This response can be explained by the fact that complex I can be bypassed by rotenone-insensitive, non–proton-pumping, type II NAD(P)H dehydrogenase activities in the mitochondrial inner membrane[42] and complex II also contributes electrons to the mETC in both cell types.

Treatment with both rotenone and TTFA, however, had a greater impact on ATP production compared to single inhibitor treatments, particularly in GCs (Fig. 3e, f). The inhibition of mitochondrial ATP synthase by oligomycin dramatically lowered cytosolic MgATP$^{2-}$ levels in MCs and GCs (Fig. 3e, f). The

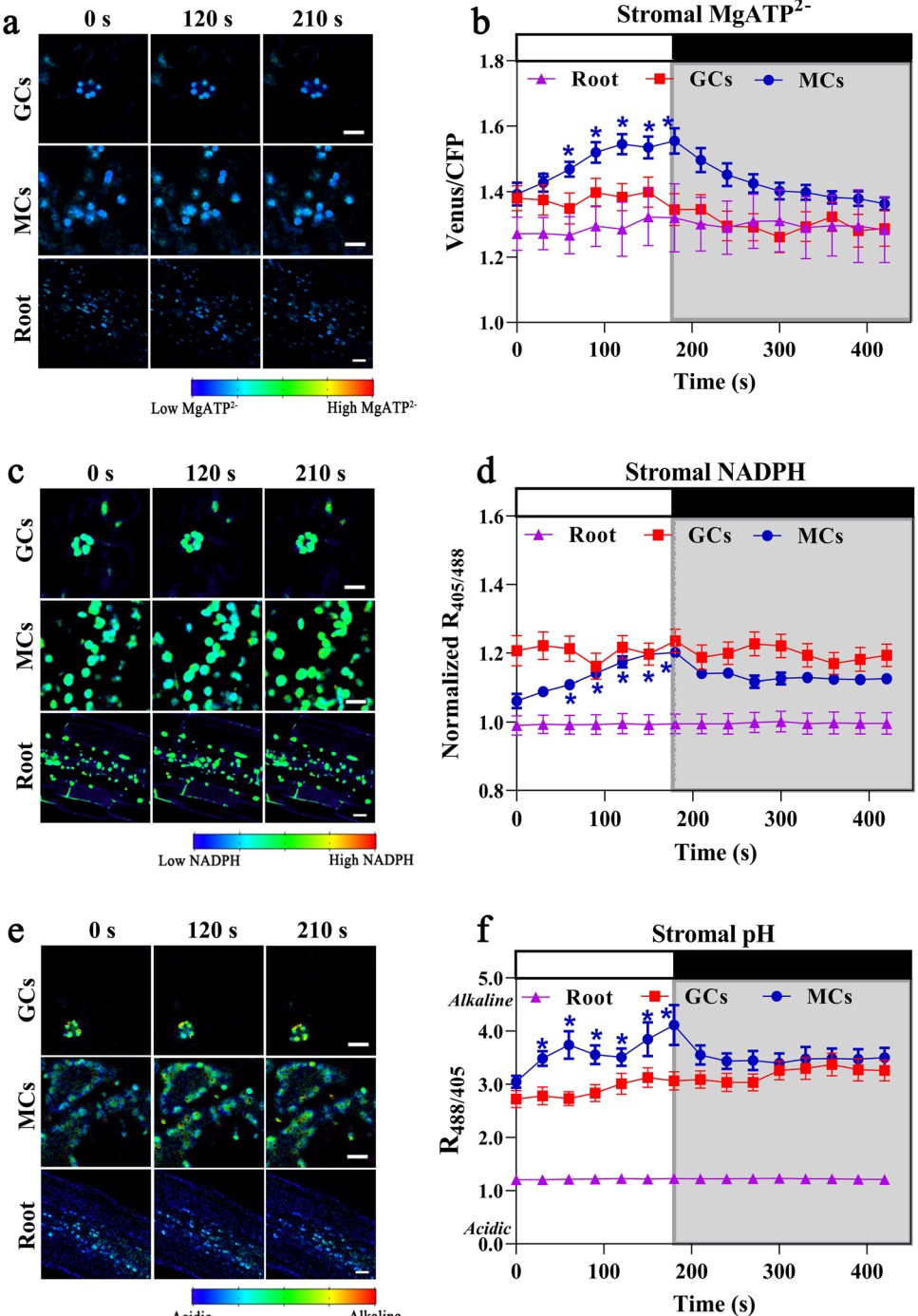

**Fig. 2 Real-time light responses of stromal MgATP²⁻, NADPH and pH sensors in 21-day-old plants.** The ratiometric shifts of **a, b** MgATP²⁻ sensor AT1.03 ($p$-values of panel b: 0 s vs. 60 s = 0.007, 0 s vs. 90 s = 0.001, 0 s vs. 120 s = 2.5 × 10⁻⁴, 0 s vs. 150 s = 1.9 × 10⁻⁴, and 0 s vs. 180 s = 4.3 × 10⁻⁴), **c, d** NADPH sensor iNAP4 ($p$-values of panel d: 0 s vs. 60 s = 0.015, 0 s vs. 90 s = 0.001, 0 s vs. 120 s = 0.001, 0 s vs. 150 s = 0.001, and 0 s vs. 180 s = 0.006), **e, f** pH sensor cpYFP ($p$-values of panel f: 0 s vs. 30 s = 0.012, 0 s vs. 60 s = 0.036, 0 s vs. 90 s = 0.021, 0 s vs. 120 s = 0.015, 0 s vs. 150 s = 0.041, and 0 s vs. 180 s = 0.032) in plastid stroma of mesophyll cells (MCs), guard cells (GCs) and roots in response to white light illumination at 216 µmol m⁻² s⁻¹ (30-s intervals for 180 s) are presented. Asterisks indicate significant statistical differences (*$P < 0.05$) between the data points collected during illumination (30–180 s) and the data point collected before illumination (0 s), as determined by paired $t$-test, two-tailed ($n = 5$; mean ± SEM). The white and black bars at the top indicate the light and dark period, respectively. Scale bars, 10 µm. All iNAP results presented were normalized with stromal iNAPc. Ratio images are presented in pseudo-color, where red corresponds to high MgATP²⁻ and NADPH levels and alkaline pH. Source data are provided as a source data file.

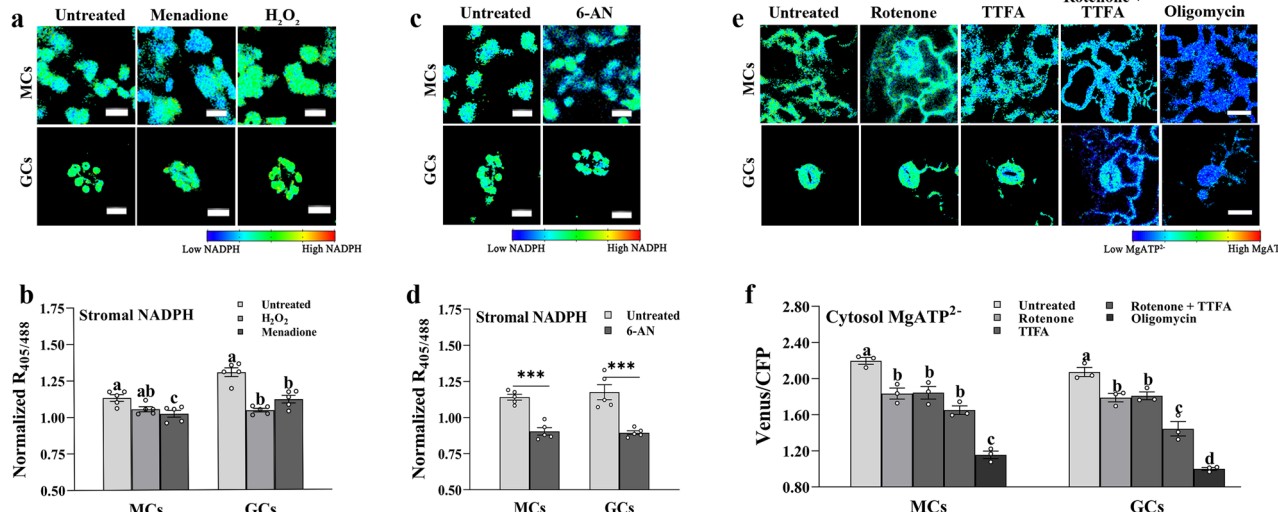

**Fig. 3 Effects of various inhibitors on stromal NADPH and cytosolic ATP. a, b** Effect of $H_2O_2$ and menadione on guard cells (GCs) and mesophyll cells (MCs) expressing the NADPH sensor TKTP iNAP4 in the stroma (one-way ANOVA with Tukey's HSD test at $P < 0.05$; $n = 5$; mean ± SEM; $p$-values of panel b (MCs): untreated vs. $H_2O_2 = 0.055$, and untreated vs. menadione $= 0.008$; $p$-values of panel b (GCs): untreated vs. $H_2O_2 = 0.2 \times 10^{-4}$, and untreated vs. menadione $= 0.4 \times 10^{-3}$). Different letters indicate significant statistical differences. Scale bars, 10 μm. **c, d** Effect of 5 mM 6-aminonicotinamide (6-AN) treatment on GCs and MCs expressing the NADPH sensor TKTP iNAP4 in the stroma. Cells were dark-adapted for 2 h prior to treatment with 6-AN (unpaired $t$-tests, two-tailed at ***$P < 0.001$; $n = 5$; mean ± SEM; $p$-value of panel d (MCs): untreated vs. 6-AN $= 0.9 \times 10^{-4}$; $p$-value of panel d (GCs): untreated vs. 6-AN $= 0.001$). Asterisks indicate statistically significant differences from untreated control. Scale bars, 10 μm. **e, f** Changes in cytosolic ATP levels upon treatment with 0.05 mM rotenone, 0.1 mM thenoyltrifluoroacetone (TTFA), 0.01 mM oligomycin or 0.05 mM rotenone with 0.1 mM TTFA in GCs and MCs (one-way ANOVA with Tukey's HSD test at $P < 0.05$; $n = 3$; mean ± SEM, $p$-values of panel f (MCs): untreated vs. rotenone $= 0.005$, untreated vs. TTFA $= 0.006$, untreated vs. TTFA + rotenone $= 0.2 \times 10^{-3}$, and untreated vs. oligomycin $= 5.7 \times 10^{-7}$; $p$-values of panel f (GCs): untreated vs. rotenone $= 0.019$, untreated vs. TTFA $= 0.03$, untreated vs. TTFA + rotenone $= 0.4 \times 10^{-4}$, and untreated vs. oligomycin $= 3.1 \times 10^{-7}$). Different letters indicate significant statistical differences. Scale bars, 20 μm. Unless stated otherwise, all treated seedlings were incubated in the dark for 1 h before imaging. Ratios of the Venus/CFP and raw iNAP4 ratio before normalization are represented in pseudo-color images, where high ratios (red) correspond to high NADPH level and high MgATP$^{2-}$. Source data are provided as a source data file.

reduction in cytosolic ATP levels in response to treatment with chemical inhibitors of mETC demonstrates that mitochondria are a major source of cytosolic ATP of MCs and, particularly, GCs of mature leaves[43].

During stomatal opening, cytosolic ATP is used to energize $H^+$ extrusion at the PM by $H^+$-ATPase[37]. It is unclear, however, whether cytosolic ATP is provided solely by mitochondria or also by GCCs, as suggested in some reports[14,15,36]. Here, we show that depletion of cytosolic ATP in GCs and MCs following oligomycin treatment abolished alkalinization of cytosolic pH in response to light, independently of the time of day (Supplementary Fig. 3a–d). Considering that light-induced alkalinization of the cytosol is a process that in GCs is intrinsic to the activation of proton pumping at PM, we conclude that $H^+$-ATPase activity (and PM $K^+$ channel activity) mainly relies on cytosolic ATP provided by mitochondria.

**GCCs import cytosolic ATP via the plastidial ATP/ADP translocator NTT1.** Besides supporting the activity of the PM $H^+$-ATPase, we hypothesized that cytosolic ATP provided by mitochondria may also be taken up by GCCs to fuel the energy-consuming processes taking place in chloroplasts.

Arabidopsis possesses two isoforms of plastidial ATP/ADP translocator, NUCLEOTIDE TRANSPORTER NTT1 and NTT2. Both proteins are localized at the inner plastid envelope membrane and mediate the exchange of ATP and ADP in antiport mode[44,45]. NTTs are expressed in developing mesophyll chloroplasts to allow the import of cytosolic ATP into the stroma to supply energy for chloroplast biogenesis[38,45,46]. By contrast, NTTs are downregulated in mature MCCs. As a result, MCCs of

mature leaves are not able to import ATP, thereby preventing the drain of cytosolic ATP by chloroplasts in the dark[38,45,46].

RT-qPCR analysis of GC-enriched epidermal peels revealed that both *NTT1* and *NTT2* genes were highly and preferentially expressed in GCs of Arabidopsis mature leaves (Fig. 4a). *NTT1* transcript levels were 36-fold higher in GCs than in whole leaves, whereas the *NTT2* gene was up-regulated only 4-fold (Fig. 4a). Promoter-β-glucuronidase (GUS) fusion analyses consistently showed stronger GUS activity in GCs of plants transformed with the *NTT1-promoter::GUS* than with the *NTT2-promoter::GUS* construct (Fig. 4b, Supplementary Fig. 4a, b).

The preferential expression of NTTs in GCs of mature leaves supports our hypothesis that GCCs import cytosolic ATP and prompted us to test it directly. We isolated GCCs and MCCs from 3-week-old plants expressing the ATP sensor in the chloroplast stroma. The diameter of protoplasts and chloroplasts isolated from GCs was approximately 1/3 the diameter of those isolated from MCs (Supplementary Fig. 5a–c). The integrity of isolated chloroplasts was verified by SYTOX™ orange staining (Supplementary Fig. 6a, b)[38]. As anticipated, addition of exogenous ATP did not alter ATP content in MCCs (Fig. 4c, d)[38]. By contrast, ATP levels in GCCs increased significantly when GCCs were incubated in a buffer containing 5 mM ATP (Fig. 4c, d). These results suggest that GCCs can import ATP, primarily through the action of NTT1.

***ntt1* mutants are nearly devoid of starch in GCs.** The comparison of ATP levels in the chloroplasts and cytosol of GCs and MCs across different times of the day revealed striking differences (Fig. 1a, b; Fig. 5a). Even though cytosolic ATP levels remained significantly higher than stromal ATP levels in both GCs and

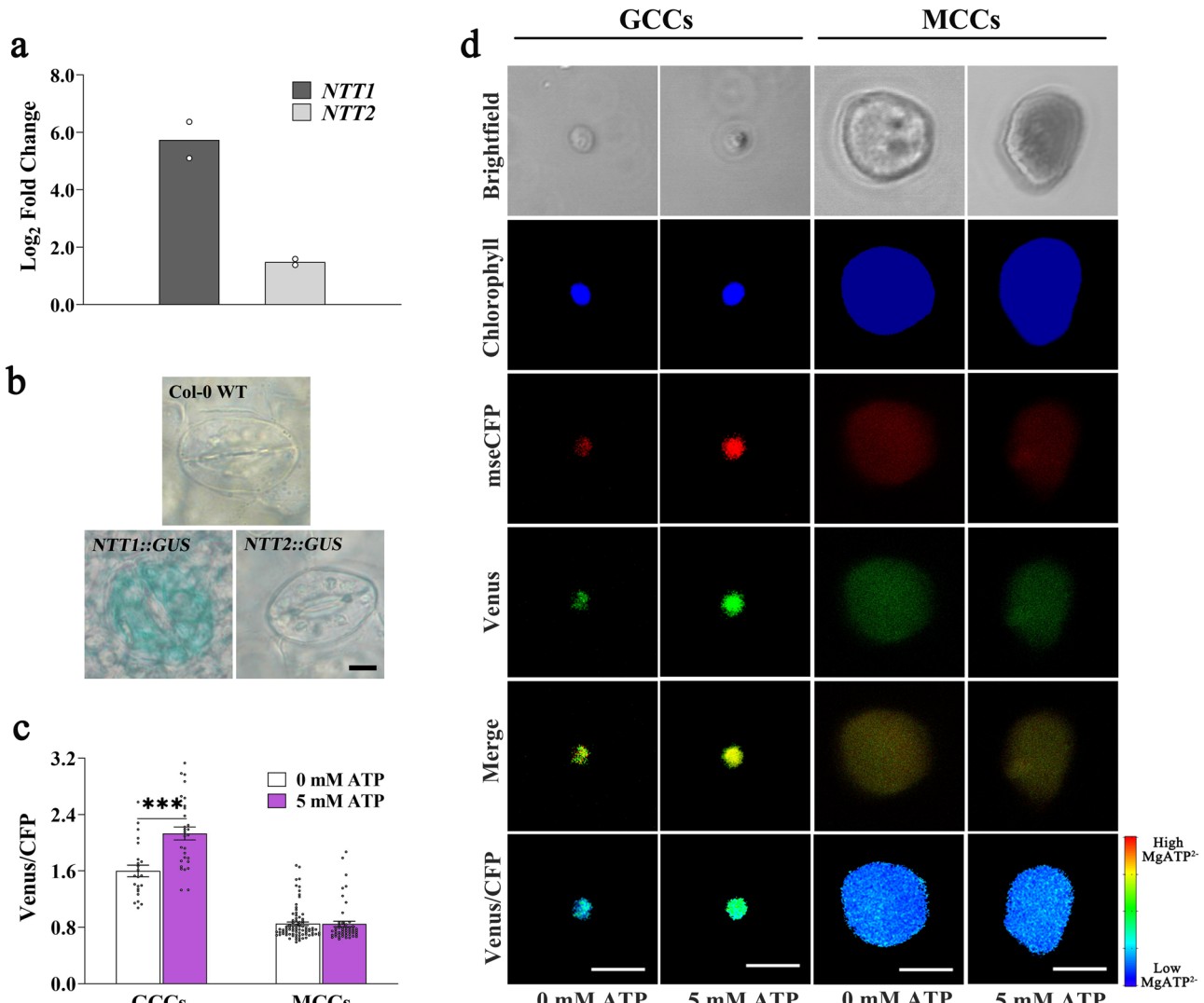

**Fig. 4 Guard cell chloroplasts import ATP via the plastidial ATP/ADP translocator NTT1. a** Relative transcript levels of the ATP/ADP antiporter *NUCLEOTIDE TRANSPORTER 1* (*NTT1*) and *2* (*NTT2*) genes in GC-enriched epidermal peels compared to leaves of wild-type plants. *ACTIN 2* was used as a housekeeping gene for normalization. Values are means of two independent experiments ± SEM. Primer sequences and efficiencies are given in Supplementary Table 1. **b** Histochemical β-glucuronidase (GUS) staining of GCs of 21-day-old leaves of plants transformed with *NTT1::promoter-GUS* or *NTT2::promoter-GUS*. This experiment was repeated three times independently. Scale bars,10 μm. **c, d** Venus/CFP ratios in isolated guard cell chloroplasts (GCCs) and mesophyll cell chloroplasts (MCCs) expressing TKTP-AT1.03 after incubation for 5 min in a buffer with or without 5 mM ATP (unpaired *t*-tests, two-tailed at ***$P < 0.001$; $n = 23, 29, 83$, and $51$, respectively; mean ± SEM; *p*-value: GCCs 0 mM ATP vs. 5 mM ATP $= 0.1 \times 10^{-3}$). Scale bars, 5 μm. Asterisks indicate significant statistical differences (***$P < 0.001$) in Venus/CFP ratios with or without exogenous ATP. Source data are provided as a source data file.

MCs at all three time points (EoN, 2 h and 8 h into the day), stromal ATP levels in MCCs gradually decreased with time, whereas ATP levels in GCCs followed a different pattern that was correlated with GC starch contents (Fig. 5a). ATP levels reached a minimum 2 h after the onset of illumination, when GC starch was scarce[12] (Fig. 5a–c). The change in stromal ATP levels might be explained by a higher rate of ATP consumption in GCCs at this time point, when GC starch is almost completely hydrolyzed and starch biosynthesis has begun.

To uncover a possible correlation between NTT activity and GC starch metabolism, we examined starch contents in GCs of the *ntt1* and *ntt2* mutants. Starch levels were significantly reduced in the mutants compared to the WT (Fig. 5b, c). However, while there was a noticeable reduction in starch content in *ntt2* GCs upon illumination, *ntt1* had GCs nearly devoid of starch at all measured time points (Fig. 5b, c). These results lend support to the idea that

ATP transport from the cytosol through the NTT transporters, particularly NTT1, is critical for starch metabolism in GCCs.

**Stomatal opening is impaired in *ntt1* mutant.** Using an infrared gas analyzer, we measured changes in stomatal conductance (*g*s) in response to light. Stomatal opening kinetics in *ntt2* were comparable to that of WT, with no statistically significant differences (Fig. 5d, Supplementary Fig. 7a, b). By contrast, *ntt1* mutant displayed strongly impaired stomatal opening responses upon illumination, with slow *g*s kinetics and a reduced amplitude compared to both WT and the *ntt2* mutant (Fig. 5d, Supplementary Fig. 7a, b). The alterations in stomatal opening kinetics slightly reduced *ntt1* photosynthetic assimilation (*A*), although the differences in *A* between the *ntt1* mutant and the WT were not statistically significant (Fig. 5e, Supplementary Fig. 7c). The *g*s kinetics of *ntt* mutants reflect the gene expression data (Fig. 4a, b)

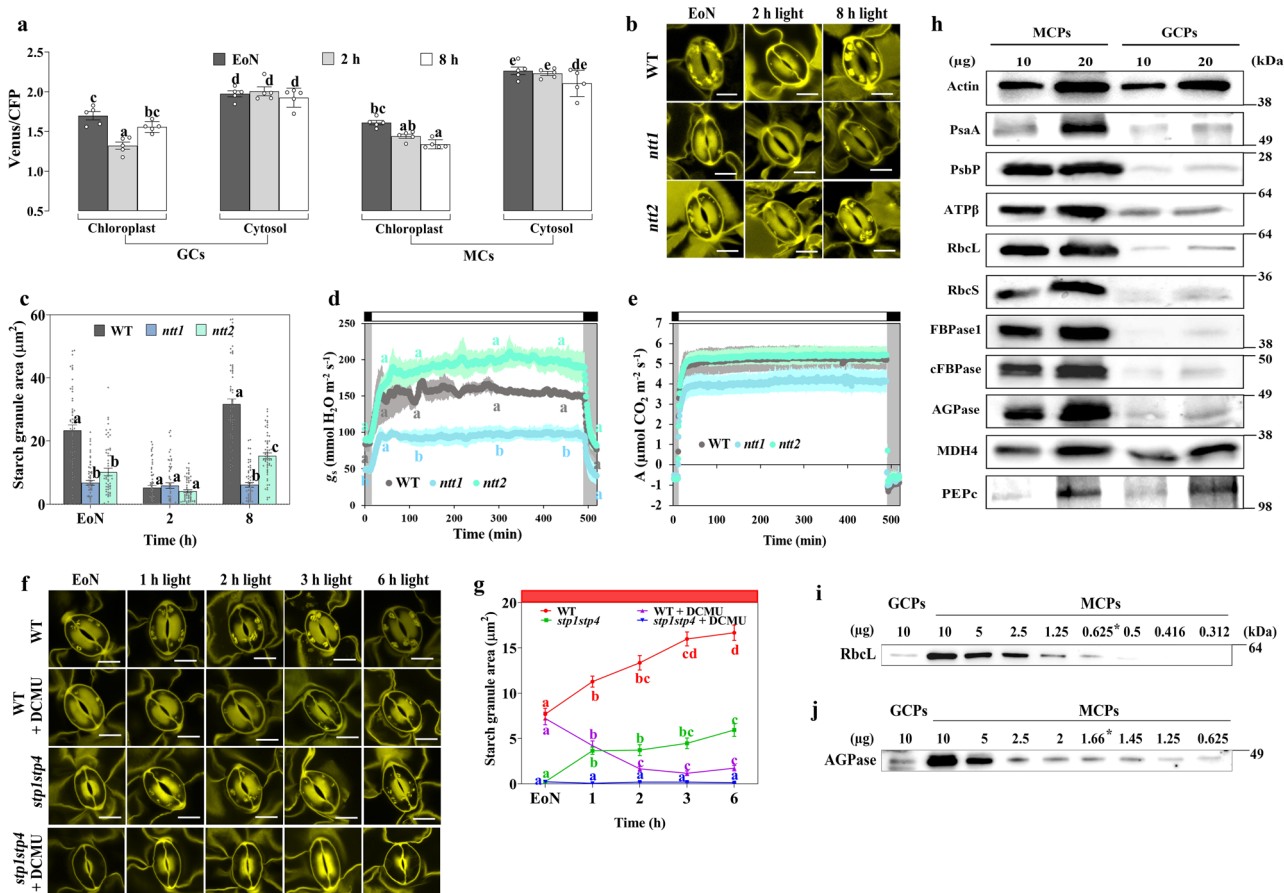

**Fig. 5 NTT is important for GC starch synthesis and stomatal opening. a** Basal MgATP$^{2-}$ Venus/CFP ratio in cytosolic and plastid stroma of guard cells (GCs) and mesophyll cells (MCs) of leaves of 20- to 22-day-old plants collected at the end of night (EoN; 0 h) or after 2 h or 8 h of white light illumination. Different letters indicate significant statistical differences analysed by one-way ANOVA and Tukey's HSD test ($P < 0.05$; $n = 5$; mean ± SEM). Exact $p$-values for panel a experiments are provided in the source data file. **b**, **c** GC starch content of wild-type, $ntt1$, and $ntt2$ plants. Representative confocal laser microscopy images of propidium iodide-stained GC starch granules. Scale bars, 10 µm. Starch granule area is given in µm$^2$. Each GC starch value represents the mean ± SEM of five biological replicates of more than 30 individual GC pairs for each group. Different letters indicate statistically significant differences among time points for the given genotype for $P < 0.05$ determined by one-way ANOVA with post hoc Tukey's test; $p$-values of panel c at EoN: WT vs. $ntt1 = 5.1 \times 10^{-9}$, and WT vs. $ntt2 = 5.1 \times 10^{-9}$; at 8 h: WT vs. $ntt1 = 5.1 \times 10^{-9}$, WT vs. $ntt2 = 5.1 \times 10^{-9}$, and $ntt1$ vs. $ntt2 = 8.0 \times 10^{-7}$. **d** Whole-plant recordings of changes in stomatal conductance ($g_s$) of wild-type, $ntt1$ and $ntt2$ plants in response to a shift from dark to light and from light to dark after 8 h of illumination at 150 µmol m$^{-2}$ s$^{-1}$ (one-way ANOVA with Tukey's HSD test at $P < 0.05$; $n = 3$ per genotype; mean ± SEM; $p$-values of panel d at 5 min: WT vs. $ntt1 = 0.046$, $ntt1$ vs. $ntt2 = 0.023$; at 120 min: WT vs. $ntt1 = 0.018$, and $ntt1$ vs. $ntt2 = 0.003$; at 300 min: WT vs. $ntt1 = 0.023$, and $ntt1$ vs. $ntt2 = 0.002$; at 450 min: WT vs. $ntt1 = 0.041$, and $ntt1$ vs. $ntt2 = 0.004$). Letters indicate significant statistical differences between genotypes for the given time points. **e** Whole-plant recordings of changes in CO$_2$ assimilation ($A$) of wild-type, $ntt1$, and $ntt2$ plants in response to a shift from dark to light and from light to dark after 8 h of illumination at 150 µmol m$^{-2}$ s$^{-1}$. **f**, **g** Starch contents of isolated GCs of wild-type and $stp1stp4$ plants illuminated for 1 h, 2 h, 3 h and 6 h with 300 µmol m$^{-2}$ s$^{-1}$ red light (RL) in the presence or absence of 10 µM 3-(3,4-dichlorophenyl)-1,1-dimethylurea (DCMU) starting at the end of the night (EoN). The isolated GCs were dark-adapted for 1 h before illumination and inhibitor treatment. Each value represents mean ± SEM of four biological replicates of more than 110 individual GCs obtained from three (control) and two (DCMU treatment) independent experiments. Different letters indicate statistically significant differences among time points for the given genotype for $P < 0.05$ determined by one-way ANOVA with post hoc Tukey's test. Scale bars, 10 µm. Exact $p$-values for panel g experiments are provided in the source data file. **h** Immunoblot analysis of proteins extracted from guard cell protoplasts (GCPs) and mesophyll cell protoplasts (MCPs) using antibodies specific for actin, PSI-A core protein of photosystem I (PsaA), 23 kDa protein of the oxygen-evolving complex of PSII (PsbP), beta subunit of ATP synthase (ATPβ), RubisCO large subunit (RbcL), RubisCO small subunit (RbcS), chloroplastic fructose-1,6-bisphosphatase 1 (FBPase1), cytosolic fructose-1,6-bisphosphatase (cFBPase), ADP-glucose pyrophosphorylase (AGPase), malate dehydrogenase 4 (MDH4) and phospho*enol*pyruvate carboxylase (PEPc). **i**, **j** Titration of the relative amounts of RbcL and AGPase in GCPs by serial dilution of MCP total proteins, using anti-RBcL and anti-AGPase antibodies. This experiment was repeated two times independently. Relative quantification of the bands was performed with the UVITEC Alliance software. Asterisks (*) indicate equivalent protein band intensities between MCPs and GCPs. Source data are provided as a source data file.

and the GC starch turnover profiles (Fig. 5b, c), showing that NTT1 is the major isoform of NTTs in GCs.

Interestingly, when the plants were subjected to a shift from light to darkness, $ntt1$ mutants again showed impaired $g_s$ responses, indicating slow and incomplete stomatal closing compared to WT and the $ntt2$ mutant (Supplementary Fig. 7d).

Previous studies reported that starch biosynthesis in GCs is involved in stomatal closing, where starch would serve as a sink for metabolites previously stored in the vacuole, which then need to be removed to reduce cell turgor[47,48]. Our findings are in line with this idea and highlight the importance of gluconeogenesis and ATP import to GCCs to promote the conversion back to

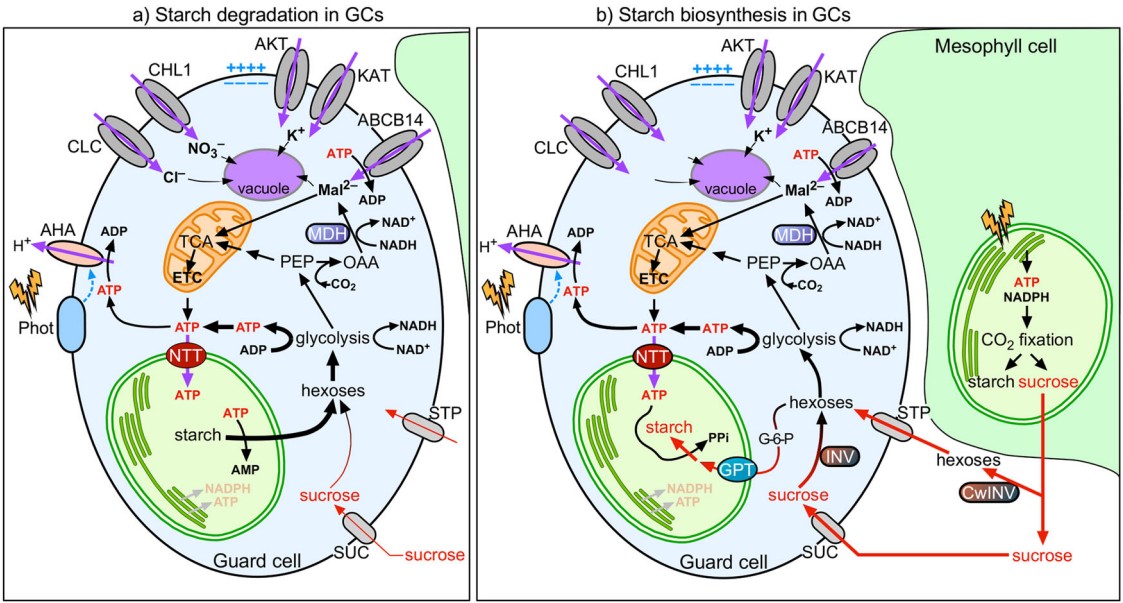

**Fig. 6 Model of the coordination of stomatal function with starch and malate metabolism in guard cells. a** Guard cell (GC) starch is broken down at the initial phase of illumination. At dawn, the GC plasma membrane-associated photoreceptor kinases PHOT1 and PHOT2 are activated by low-irradiance blue light to induce stomatal opening[6]. In parallel, GC starch is mobilized to sustain sugar homoeostasis for stomatal opening[13]. Sugars are also an energy source for GC mitochondria to generate ATP and reducing equivalents[35]. Through glycolysis, sugars are converted to PEP, a 3C compound, which can fix one $CO_2$ molecule and be further converted into OAA, a 4C compound, via the action of PEPc (phospho*enol*pyruvate carboxylase). OAA is then further reduced to $Mal^{2-}$ by MDH[35]. $Mal^{2-}$ is imported to the vacuole to contribute to GC osmoregulation and stomatal opening and acts as a counter-ion for $K^{+}$ [76]. $Mal^{2-}$ can also be directly imported from the apoplast[58,77]. At the same time, carbon fixation and starch/sucrose biosynthesis occur in mesophyll cells. **b** Starch biosynthesis occurs in GCs after the initial phase of illumination. After complete starch degradation in GCs, stomatal opening is sustained by mesophyll cell-derived sugars[10,49]. Hexoses, derived from sucrose from neighbouring mesophyll cells through the action of cell wall invertase, are imported to GCs via the monosaccharide/$H^{+}$ cotransporters STP1 and STP4[10,78]. In GCs, hexoses are converted into glucose-6-phosphate, which is then transported into the chloroplast through Glc-6-P/Pi translocators and can be utilized for starch biosynthesis[33,79]. The imported sugars can also be metabolized through glycolysis[18] and mitochondrial respiration to generate cytosolic ATP that enters GCCs via ATP/ADP antiporter NTTs to deliver ATP for starch biosynthesis. Chloroplasts and mitochondria are represented by green and brown circles, respectively. Red arrows represent the energy source flow. **ADP** Adenosine diphosphate; **AHA** $H^{+}$-ATPase; **AKT** Inward-rectifying $K^{+}$ channel; **AMP** Adenosine monophosphate; **ATP** Adenosine triphosphate; **CHL1** Dual-affinity nitrate transporter; **CLC** Chloride channel of the CLC gene family; **CwINV** Cell wall invertase; **G6P** Glucose-6-phosphate; **GPT** Glucose-6-phosphate/Pi translocator; **$H^{+}$** Proton; **$K^{+}$** Potassium; **KAT** Inward-rectifying $K^{+}$ channel; **MDH** Malate dehydrogenase; **NTT** Nucleotide transport protein; **OAA** Oxaloacetate; **PEP** Phospho*enol*pyruvate; **Phot** Phototropin; **STP** Sugar Transport Protein; **SUC** Sucrose /$H^{+}$ cotransporter; **V** Vacuole. Imported ions and sugars are stored in the vacuole.

starch of organic acids and sugars previously accumulated by the GCs.

Taken together, these data suggest that NTT1 is required for proper stomatal movements, likely by providing ATP to the GCCs to energize starch turnover and other associated metabolic processes (Fig. 6).

**DCMU inhibits starch synthesis in isolated GCs.** Besides energy, starch biosynthesis requires the supply of carbohydrate precursors. There are three possible carbon sources for GC starch synthesis: photoassimilates from GC photosynthesis, sugars imported from the mesophyll, or organic acids previously accumulated within the GCs, which can be converted back to starch via gluconeogenesis[33,34,49–52]. While it is well established that the coordinated actions of the photosystems and CBB cycle deliver the precursors required for starch biosynthesis in MCCs[53], it is unclear to what extent the CBB cycle is active in GCCs and what its relative contribution is to the pool of accumulated starch[33,34].

Red light (RL) promotes efficient starch synthesis in GCs, likely through induction of photosynthetic $CO_2$ fixation[12,13]. By comparing the ability to accumulate starch in response to RL of GCs of intact leaves and GCs in isolated epidermal peels in which there is no connection with the mesophyll, we previously demonstrated that GCs mostly rely on mesophyll-derived sugars for starch

biosynthesis[13]. We further reported that GCs of the Arabidopsis *stp1stp4* mutants, lacking the PM monosaccharide-$H^{+}$ symporters SUGAR TRANSPORT PROTEIN 1 and 4 (STP1 and STP4), have almost undetectable amounts of glucose at dawn and are nearly devoid of starch[10]. Thus, STP1 and STP4 mediate the import of mesophyll-derived glucose to GCs, which is used as a main precursor for GC starch biosynthesis at dawn[10]. While these findings highlight the predominant role of mesophyll photosynthesis, they do not exclude the possibility that $CO_2$ fixation in GCCs may contribute to starch accumulation. However, direct evidence connecting GC photosynthesis with starch metabolism is still lacking.

Here, we examined changes in GC starch contents in response to RL illumination in isolated GCs of WT and *stp1stp4* mutant plants. As observed previously, isolated WT GCs accumulated starch in response to a 6-h RL treatment (Fig. 5f, g)[13]. Isolated GCs of *stp1stp4* mutants also accumulated starch, though to a substantially reduced level compared to WT (Fig. 5f, g). Based on our previous results[10,13], we assume that isolated GCs of WT plants synthesized starch mostly using imported hexose sugars that were present in the epidermal peel apoplast at the beginning of the assay. However, the fact that *stp1stp4* isolated GCs were also able to accumulate starch, despite their defective PM sugar transport system, made us wonder whether $CO_2$ fixation in GCCs directly contributed to starch synthesis under RL.

To test this hypothesis, we applied 10 µM 3-(3,4-dichlorophe-nyl)-1,1-dimethylurea (DCMU), an inhibitor of photosystem II (PSII), to the isolated GCs prior to illumination with RL. Starch levels significantly decreased in WT GCs in response to DCMU, and GCs were almost fully starch-depleted after 2 h of treatment (Fig. 5f, g). In contrast to WT, starch levels in DCMU-treated *stp1stp4* isolated GCs immediately dropped to zero and remained low for the duration of the experiment (Fig. 5f, g). The response to DCMU of WT GCs, and particularly *stp1stp4* mutants, strongly suggests that GC photosynthesis, stimulated by RL illumination, provided isolated GCs with carbon precursors for starch accumulation. We therefore conclude that GC photosynthesis at least partly contributes to starch synthesis in GCCs.

**Enzymes for phototropic $CO_2$ fixation are present in small amounts in Arabidopsis GCs.** To further investigate the con-tribution of GC photosynthesis to $CO_2$ fixation in GCs, we determined the abundance of proteins involved in photosynthesis and anaplerotic $CO_2$ fixation in GCPs and MCPs (Fig. 5h–j). Given that GCPs and MCPs greatly differ in size (Supplemental Fig. 5a, c), extracted proteins were quantified and equalized before loading, and actin was used as an internal standard for immunoblot analysis (Fig. 5h).

Compared to MCPs, GCPs contained smaller but still detectable amounts of the photosynthetic proteins PsaA (PSI protein), PsbP (PSII protein), RubisCO (RbcL and RbcS), the β subunit of the ATP synthase (ATPβ), and the chloroplastic fructose-1,6-bisphosphatase (FBPase1, a CBB cycle enzyme) (Fig. 5h, i). Immunoblotting titrations further revealed that Arabidopsis GCPs contained larger amounts of ADP-glucose pyrophosphorylase (AGPase, a critical enzyme of starch bio-synthesis) than RubisCO (Fig. 5i, j). These results support our observation that GC photosynthesis is at least partly functional in Arabidopsis GCCs and suggest that AGPase is an abundant enzyme in GCCs.

We also detected very small amounts of cytosolic fructose-1,6-bisphosphatase (cFBPase, Fig. 5h). cFBPase converts fructose-1,6-bisphosphate to fructose-6-phosphate, which is the reverse reaction catalysed by phosphofructokinase in glycolysis. The low levels of cFBPase are in line with the idea that GCs have a high glycolytic activity in the light[16–18] and suggest that GC central metabolism is adapted to dissimilate rather than synthesize sucrose during dark-to-light transition.

In contrast to the reduced amounts of photosynthetic proteins, we found that proteins involved in anaplerotic $CO_2$ fixation, such as NAD-malate dehydrogenase (MDH4) and particularly PEPc, were present in larger amounts in GCPs than in MCPs (Fig. 5h). These data suggest that although photosynthesis is active, Arabidopsis GCs mainly assimilate $CO_2$ via PEPc.

## Discussion

It has long been suggested that the energy required for stomatal opening could come from oxidative phosphorylation. This idea is mostly based on indirect evidence, including the occurrence of numerous mitochondria and higher respiratory rates in GCPs compared to MCPs[20,21], the reduction of stomatal opening by treatment of epidermal peels with respiratory inhibitors, the ability of stomata to open to a remarkable extent even in darkness[22], and the inability of BL-induced stomatal opening at low $O_2$ level[54].

Regardless, some other reports suggested that photopho-sphorylation in GCCs is essential to provide ATP to the cytosol for light-induced stomatal opening[14,15,28,36]. Evidence for GC photosynthesis and ATP export to the cytosol is also circum-stantial, mostly based on measurements of stomatal opening

under various combinations of BL-RL stimuli in presence or absence of DCMU[14,36,55] or on quantification of ATP content in GCs using luciferase assays[14,15,28]. It is assumed that ATP would be indirectly transferred to the cytosol through a phosphoglyce-rate (PGA)/dihydroxyacetone phosphate (DHAP) shuttle across the chloroplast envelope[28,56], but the protein has not yet been identified in GCs.

Our approach to use ratiometric fluorescence protein sensors bypassed the complexity of in vitro measurements of absolute amounts of ATP and NADPH, which have very low abundance in GCs (perhaps in the order of pmol[28]) and a very short half-life. The non-destructive nature of *in planta* imaging further allowed the analysis of real-time dynamic changes in energy metabolism in living GCs at the subcellular level, something that was pre-viously impossible.

Tracking changes in FRET ratios in GCs expressing the AT1.03 sensor in the plastid stroma demonstrated not only that the ability of GCCs to produce ATP in response to light is neg-ligible (Fig. 1b, Fig. 2b), but also that oligomycin treatment drastically reduced cytosolic ATP levels (Fig. 3e, f). We also observed that depletion of cytosolic ATP by oligomycin abolished light-induced alkalinization of cytosolic pH (Supplementary Fig. 3d), a process associated with the activation of PM H$^+$ pumping. These findings define mitochondria as the primary source of ATP for GCs and show that PM H$^+$-ATPase activity upon illumination is mainly fuelled by ATP produced via the tricarboxylic acid (TCA) cycle and oxidative phosphorylation.

In line with this conclusion, recent $^{13}C$ feeding experiments in GCs showed that $^{13}C$-enrichment into Mal$^{2-}$ decreased within 60 min of illumination with a concomitant increase in the $^{13}C$ label in some intermediates of the TCA cycle[17,57]. Thus, Mal$^{2-}$ imported from the apoplast[58] or produced within GCs from PEP (an intermediate of glycolysis) is a major substrate for the TCA cycle, contributing to ATP production via oxidative phosphor-ylation. The need for large amounts of Mal$^{2-}$ in GCs at dawn may explain the activation of glycolysis in the light[16]. Dissimilation of sucrose and starch within GCs during dark-to-light transition[12,17,18] could represent a means to quickly provide carbon skeletons for glycolysis to meet the high energetic demand of ion transport. GCs also catabolize stored triacylglycerols at dawn via β-oxidation in a BL-dependent manner[59,60]. β-oxidation in peroxisomes generates NADH and acetyl-CoA. NADH is recycled to NAD$^+$ by the peroxisomal MDH, ultimately releasing Mal$^{2-}$ for import to mitochondria, where it can provide a source of NADH to the TCA cycle. These observations suggest that GCs have adapted their metabolism towards the production of Mal$^{2-}$ and ATP to sustain stomatal opening.

Despite low photosynthetic activity, we found that GCCs have considerable levels of both ATP and NADPH (Fig. 1a–d, Fig. 2a–d). Our work unequivocally demonstrates that GCCs compensate for their inability to generate sufficient ATP by importing cytosolic ATP provided by mitochondria through the plastidial ATP/ADP translocator NTT1 (Fig. 4a–d, Fig. 6). A fraction of the imported ATP is directly used to energize starch synthesis, which, in turn, is essential for light-regulated stomatal movements (Fig. 5b–e, Supplementary Fig. 7a, b). The severe phenotype of the *ntt1* mutant, which is nearly devoid of starch in GCs (Fig. 5b, c), suggests that starch metabolism is in delicate balance with organic acid and energy metabolism in mitochon-dria. The exchange of metabolites between these subcellular compartments is an important channel of communication, ulti-mately coordinating the energetic and metabolic status of the cell with membrane ion transport activity.

Based on our results showing that treatment of GCs with 6-AN inhibits stromal NADPH accumulation (Fig. 3c, d), we suggest that NADPH in GCCs is produced mainly by OPPP in the dark.

It cannot be excluded that other processes, such as $Mal^{2-}$ decarboxylation by the malic enzyme[57], may also contribute to NADPH production in GCCs. Although early studies pointed to high levels of malic enzyme activities in *V. faba* epidermal tissue[61], experimental validation in Arabidopsis is still missing. Furthermore, $Mal^{2-}$ decarboxylation in the chloroplast would require the exchange of cytosolic $Mal^{2-}$ with glutamate (produced through the ferredoxin-dependent glutamate synthase) by the dicarboxylate transporter[57], a protein that has not yet been identified in GCs.

The observation that GCCs obtain ATP and NADPH via different routes from MCCs is striking and suggests that GCs behave more like a sink than a source tissue. For example, it was recently shown that similar to GCs, down-regulation of *StNTT* in potato (*Solanum tuberosum*) tuber results in considerably less starch accumulation[62]. Here, comparison of real-time light responses of stromal $MgATP^{2-}$, NADPH and pH sensors between GCs, MCs, and root cells of Arabidopsis further confirmed that GCs possess characteristics of sink tissues (Fig. 2a–f). However, GCs showed overall larger amounts of ATP and NADPH and a more alkaline stroma compared to root cells (Fig. 2b, d, f). It seems likely that GC metabolism is adapted to activate glycolysis and mitochondrial metabolism in the light to increase the rate of ATP production, possibly as a response to the low chlorophyll content found in these in cells.

Although these findings may imply a small or insignificant role of photosynthesis in GCs, recent work demonstrated that transgenic Arabidopsis plants with degraded chlorophyll specifically in GCs exhibited a deflated, thin GC phenotype and reduced stomatal conductance[63]. Hence, chlorophyll at the thylakoid membrane seems critical for GC turgor, but the way it contributes to it, through either production of sugars via the CBB cycle[32], or as some type of signalling component[64,65], remains unclear.

Inhibition of photosynthesis in isolated GCs by DCMU under RL illumination blocked starch accumulation (Fig. 5f, g). While GCs of *stp1stp4* mutants responded immediately to the treatment, WT GCs showed a delayed response and were fully starch-depleted only after 2 h of treatment (Fig. 5g). Because *stp1stp4* mutants are impaired in sugar transport at the GC PM[10], we interpret the response to DCMU as a result of a difference in their ability to import hexose sugars that were present in the epidermal peel apoplast at the beginning of the assay. It seems reasonable that starch synthesis stopped in isolated WT GCs once they exhausted imported carbon precursors. In the case of *stp1stp4*, however, no such apoplastic sugars were available for GCCs at the time of DCMU treatment, thereby causing immediate inhibition of starch synthesis. The use of *stp1stp4* mutants allowed us to discriminate between the contribution of imported apoplastic sugars and GC photosynthesis-derived sugars to starch synthesis. Our results provide direct evidence that GC photosynthesis is required for proper starch accumulation in GCCs and are in line with a previous report showing that starch levels were significantly reduced in GCCs lacking chlorophyll[63].

Further support for the conclusion that photosynthesis is active in Arabidopsis GCCs comes from our Western blot analyses, demonstrating that several photosynthetic proteins, including RubisCO, and the CBB cycle enzyme FBPase1 were present at detectable levels in GCPs (Fig. 5h, i). Our data corroborate earlier findings from several other C3 species, in which RubisCO has been localized to GCs using in situ immunofluorescence[26].

That said, cytosolic MDH4 and especially PEPc were highly abundant in Arabidopsis GCs and present in larger amounts compared to MCs (Fig. 5h), in agreement with the overall idea that anaplerotic $CO_2$ fixation is the main pathway of carbon assimilation in GCs[17,50,52,57,66]. For example, early $^{14}C$-labelling experiments showed that $CO_2$ was fixed into $Mal^{2-}$ at high rates both in the dark and the light in GCs, whereas in MCs, $^{14}C$ was enriched in 3-PGA and sucrose in the light and $Mal^{2-}$ only in the dark[50]. Furthermore, recent $^{13}C$-feeding experiments demonstrated a faster $^{13}C$ enrichment in $Mal^{2-}$ in isolated epidermal fragments compared with the whole rosette of Arabidopsis[17,52,67]. Fixation of $CO_2$ via PEP carboxylation ultimately leads to the formation of $Mal^{2-}$, further highlighting the central role of this metabolite for GCs.

In conclusion, our work demonstrates that GCCs and MCCs greatly differ in the way they obtain energy and carbon skeletons. GC photosynthesis is poorly active, and mitochondria are the major source of ATP for GCs. Unlike MCs, GC metabolism mainly favours sucrose and starch degradation upon transition from dark to light. Starch-derived sugars and sugars imported from the mesophyll maintain the cytoplasmic sugar pool needed for the activation of glycolysis and mitochondrial metabolism in the light. NTTs at the inner plastid envelope membrane facilitate the import of mitochondrial-derived ATP into GCCs, sustaining starch turnover. $Mal^{2-}$ seems to be a central metabolite for GCs and is produced in high amounts to meet the energy demand, but also to provide counterions and osmotica to promote stomatal opening. To integrate the findings of this work with that of previous studies, we propose a model on how ATP, starch, and $Mal^{2-}$ metabolism coordinate with stomatal function (Fig. 6).

## Methods

**Plant materials.** The binary vectors pH2GW7-C-cpYFP (cytosolic pH sensor), pH2GW7-TKTP-cpYFP (plastid stromal pH sensor), pH2GW7-C-AT1.03 (cytosolic $MgATP^{2-}$ sensor), pEarleyGate100-TKTP-AT1.03 (plastid stromal $MgATP^{2-}$ sensor), pEarleyGate100-TKTP-iNAP4 (plastid stromal NADPH sensor) and pEarleyGate100-TKTP-iNAPc (control sensor for iNAP4) were introduced into wild-type (WT) Columbia (Col-0) *Arabidopsis thaliana* (Arabidopsis) plants as previously described[38,39,68]. The transfer DNA (T-DNA) insertion lines SALK_083518c (*ntt1*) and SALK_031126c (*ntt2*) were obtained from the Arabidopsis Biological Resource Center. The *stp1stp4* mutant was described previously[10]. Arabidopsis plants were grown in soil in a growth chamber under a photoperiod of 12 h light (150 μmol photon $m^{-2}$ $s^{-1}$) at 22 °C and 12 h dark at 18 °C. All experiments were performed with 20- to 22-day-old plants unless stated otherwise.

**RNA isolation and quantitative PCR analysis.** Total RNA was extracted as reported in Flütsch et al.[13] The leaves from three entire rosettes (three biological replicates) were collected and frozen in liquid nitrogen before RNA extraction. For total RNA from guard cell-enriched epidermal peels, the middle veins of 12 plants were excised per biological replicate, blended (Philips, ProBlend Avance), and then passed through a 200-μm nylon mesh (Sefar). Guard cell-enriched epidermal peels were patted dry, collected and frozen in liquid nitrogen. We used three biological replicates for each experiment.

Total RNA was extracted using the SV Total RNA Isolation Kit (Promega) following the manufacturer's instructions. We used 1 μg total RNA for first-strand cDNA synthesis with the M-MLV Reverse Transcriptase, RNase H Minus Point Mutant, and oligo(dT)$_{15}$ primer (Promega). Transcript levels were determined by RT-qPCR with the SYBR green master mix (Applied Biosystems) on a 7500 Fast Real-Time PCR System (Applied Biosystems). RT-qPCR was performed in technical triplicates. Transcript levels were calculated according to the comparative $C_T$ method (Livak and Schmitgen, 2001) and normalized against the expression of *ACTIN2* (*ACT2*; At3g18780). Error calculations were done according to Applied Biosystems guidelines (http://assets.thermofisher.com/TFS-Assets/LSG/manuals/cms_042380.pdf). Primers used for RT-qPCR are reported in Supplementary Table 1.

**Generation of NTT1::promoter-GUS and NTT2::promoter-GUS Plants.** Primers *NTT1::Promoter*_F, *NTT1::Promoter*_R, *NTT2::Promoter*_F, *NTT2::Promoter*_R were used to amplify about 1.5 kb promoter regions upstream of *NTT1* and *NTT2* genes from genomic DNA by PCR[45] (Supplementary Table 2). To generate the *promoter*-GUS constructs, the promoter PCR products were cloned upstream the *GUS* gene in the binary vector *pBI121* which 35 S promoter had been removed by HindIII and XmaI restriction enzymes. Both constructs were sequenced to confirm the identity of the cloned insert.

**Histochemical localization of GUS.** The 4th and 5th leaves of 21-day old plants were excised and fixed with 90% (v/v) acetone at 4 °C for 20 min. Subsequently, the samples were washed twice with GUS washing buffer supplemented with 25 mM $NaH_2PO_4 \cdot H_2O$ (pH 7.2), 25 mM $Na_2HPO_4 \cdot 12H_2O$ (pH 7.2), 0.1% (v/v) Triton X-

100, 2 mM K$_4$[Fe(CN)$_6$]•3H$_2$O, 2 mM K$_3$Fe(CN)$_6$, and 5 mM EDTA-2Na. 2 mM X-Gluc (Sigma 203783) dissolved in GUS washing buffer was added and vacuum infiltrated for 20 min on ice to stain the leaves. After overnight incubation at 37 °C in the dark, GUS stainings were examined under a light microscope.

**Isolation of guard cell and mesophyll cell chloroplasts**. Protoplasts were isolated from 20- to 22-day-old plants as previously described[69]. Mesophyll protoplasts were isolated using the tape-Arabidopsis sandwich method[70]. Guard cell chloroplasts were then isolated using a syringe filled with protoplasts and pressed through a piece of 1-µm or 5-µm nylon mesh for guard cell protoplasts or mesophyll protoplasts, respectively. After isolation, chloroplasts were washed and resuspended in a buffer consisting of 300 mM sucrose, 50 mM HEPES-KOH pH 7.5, 10 mM KCl, 10 mM NaCl, 5 mM MgCl$_2$, and 0.1% (w/v) BSA. Chloroplasts were incubated in resuspension buffer with or without 5 mM ATP for 5 min at room temperature before imaging[38]. We detected leaky chloroplasts with 25 nM SYTOX$^{TM}$ orange nucleic acid stain, with excitation at 543 nm and emission collected from 565 nm to 604 nm. The presence of SYTOX orange lowers the FRET ratio of the ATP sensor AT1.03, as the emission of AT1.03 (526–545 nm) overlaps with the excitation range of SYTOX and some emitted light is absorbed by the dye.

**Immunoblot analysis**. Mesophyll cell and guard cell protoplasts were isolated as above. Protoplasts were lysed in lysis buffer on ice (25 mM Tris-HCl pH 7.5, 150 mM NaCl, 1 mM EDTA, 1% (v/v) Triton X-100, 1% (w/v) complete protease inhibitor cocktail (Roche, Germany)). After 30-min incubation, the lysate was centrifuged at 16,000 g for 15 min at 4 °C to collect the soluble proteins in the supernatants. Protein concentrations were determined using the Bradford protein assay (Bio-Rad, USA). After electrophoresis on SDS-PAGE gels, proteins were transferred to Amersham Protran Supported nitrocellulose membranes (GE healthcare, Hong Kong) for immunoblotting with the following antibodies; anti-plant actin (A01050, 1:3000), anti-rubisco large subunit (RbcL) (A01110, 1:3000), and horseradish peroxidase (HRP)-conjugated anti-mouse secondary antibody, which were all obtained from Abbkine (Hong Kong), anti-AGPase (AS111739, 1:3000), anti-ATPβ (AS05085, 1:4000), anti-FBPase (AS194319, 1:5000), anti-cFBPase (AS04043, 1:1000), anti-MDH4 (AS153065, 1:1000), anti-PEPc (AS09602, 1:1000), anti-PsaA (AS06172, 1:5000), anti-PsbP (AS06142-23, 1:2000), and anti-rubisco small subunit (RbcS) (AS07259, 1:3000) antibodies, which were obtained from Agrisera (Sweden). The relative quantification of each band was executed with UVITEC Alliance software (Uvitec). Anti-MDH4 antibody was raised against the recombinant full-length maize (Zea mays) MDH4 protein, which exhibits 92% identity to the cytosolic NAD-MDH, 43% to the plastidic NAD-MDH, but no homology to the mitochondrial or the peroxisomal NAD-MDH or NADP-MDH from Arabidopsis. Sequence identity was determined by BLASTP at the Arabidopsis Information Resource (TAIR) website (https://www.arabidopsis.org/Blast/) using the full-length maize MDH4 protein sequence (UniProt: Q08062) as a query.

**Confocal imaging and image processing**. Confocal imaging of the abaxial layer of leaves was set up as previously described[71]. Leaves were collected from 20- to 22-day-old plants at end of night (EoN) or 2 h or 8 h into the day. Imaging was performed with 40× oil immersion lenses in multitrack mode using a Zeiss LSM710 NLO confocal microscope (Carl Zeiss Microscopy). After the first image was obtained, leaves were further illuminated for 3 min (216 µmol m$^{-2}$ s$^{-1}$) with a halogen lamp (HAL 100 W; Philips) of the confocal microscope before the second image was captured. Plants expressing cpYFP, iNAP4, and iNAPc were excited sequentially at 405 nm and 488 nm, and the emission signals were collected at 520 ± 16 nm. Autofluorescence was recorded from 431 nm to 469 nm. The normalized iNAP4 R$_{405/488}$ was corrected with iNAPc and calculated as previously described[72]. The ATP sensor AT1.03 was excited at 458 nm and its emission was collected from 470 nm to 507 nm (Em470–507, mseCFP image) and from 526 nm to 545 nm (Em526–545, FRET image). Plants expressing AT1.03 were also excited at 515 nm (at 0.18% of maximal laser power for all samples) to excite Venus; emission was detected from 526 nm to 545 nm (cp173Venus image). The Venus/CFP ratio was calculated by dividing the fluorescence intensity of the mseCFP image with that of the FRET image. Chlorophyll autofluorescence was also captured for all images from 629 nm to 700 nm. Ratiometric images were analysed on a pixel-by-pixel basis using x, y noise filtering. Fluorescence background subtraction was conducted based on the intensity from the dark portion of the images. Confocal images were processed with a custom MATLAB-based analysis suite[73]. All ratio representative profiles are presented in pseudo colours.

**Visualisation of starch in GCCs**. The method of starch visualisation in guard cell chloroplasts was adopted from Flütsch et al.[74]. After epidermal peels were harvested using precision tweezers from the abaxial side of the 3$^{rd}$ or 4$^{th}$ leaf of 20- to 22-day-old WT Arabidopsis plants after 0 h, 2 h, or 8 h of illumination, the peels were fixed in fixative solution (50% (v/v) methanol, 10% (v/v) acetic acid) for at least 24 h at 4 °C in the dark. The fixative solution was then removed, and the peels were washed with 1 mL of dH$_2$O by shaking the plate slowly in circular movements. Chlorophyll from epidermal peels was removed by incubation in 1 mL 80% (v/v) ethanol for 15 min. Peels were rinsed with dH$_2$O, incubated in 1 mL fixative solution for 1 h at room temperature, washed with dH$_2$O, fully covered in

1% (v/v) periodic acid solution, incubated for 30 min at room temperature and then washed with dH$_2$O. Next, peels were stained with 500 µL Schiff reagent (1.9 g sodium metabisulfite, 3 mL of 5 N HCl and 97 mL dH$_2$O) and 50 µL of 1 mg mL$^{-1}$ propidium iodide solution for 30 min at room temperature. At this stage, peels appeared pinkish. Samples were destained in dH$_2$O for 30 min at room temperature. Chloral hydrate solution (40 g chloral hydrate, 10 mL glycerol, and 20 mL dH$_2$O) was added onto microscope slides and the stained peels were gently transferred onto the microscope slides and incubated in the dark overnight. Excess chloral hydrate solution was removed using wipes before mounting. Hoyer's mounting solution (30 g gum arabic, 200 g chloral hydrate, 20 g glycerol, and 50 mL dH$_2$O) was added onto the peels and covered with a cover slip. The samples were stored at room temperature in the dark for 3 days before visualisation by confocal microscopy. The samples were excited at 488 nm and emission was collected from 610 nm to 640 nm. Starch area was quantified using the image processing software ImageJ (NIH, https://imagej.nih.gov/ij/index.html).

**Gas exchange measurements**. For whole-plant gas exchange measurements, plants were grown in a Klimaschrank (Kälte3000) under a photoperiod of 8 h light (150 µmol photon m$^{-2}$ s$^{-1}$) at 21 °C with 45% relative humidity (RH) and 16 h dark at 19 °C with 55% RH. Gas exchange measurements were carried out using a 6400 XT Infrared Gas Analyzer equipped with a 6400-18 light source and the whole-plant Arabidopsis 6400-17 chamber (LI-COR Biosciences). To prevent any water vapour and CO$_2$ diffusion from the soil, the pots were sealed with clear film. All measurements were performed at 22 °C, 45–55% RH, and 400 µg L$^{-1}$ CO$_2$. Before measurements, plants were equilibrated in darkness for 30 min until all parameters had stabilized. After the reading was constant for 10 min, an irradiance of 150 µmol m$^{-2}$ s$^{-1}$ was applied to the rosette for 8 h, followed by 30 min exposure to darkness. Measurements of net A and g$_s$ values were performed on three different plants per genotype, starting always at the same time of the diurnal cycle (EoN). Parameters were recorded every minute. The whole rosette area was determined using the software ImageJ version 1.48 (NIH USA, http://rsbweb.nih.gov/ij/). The g$_s$ and A values were normalized by subtracting the conductance values at EoN (set as 0 = initial values for g$_s$ or A) as described by Baroli et al.[75] or alternatively for stomatal closure by subtracting the conductance values at 10 min before the application of darkness (set as 0 = initial values for g$_s$ or A).

**Inhibitor treatments**. Seedlings were infiltrated for 5 min in half-strength Murashige and Skoog medium with or without the following inhibitors: 0.05 mM rotenone (mitochondrial complex I inhibitor), 0.1 mM TTFA (mitochondrial complex II inhibitor), 0.01 mM oligomycin A (mitochondrial ATP synthase inhibitor), and 5 mM 6-AN (oxidative pentose phosphate pathway inhibitor). We used 10 mM H$_2$O$_2$ or 0.03 mM menadione as oxidizing agents. Unless stated otherwise, after infiltration, all seedlings were incubated in the dark for 1 h before imaging.

**Light intensity analysis**. Plants were exposed to 216 µmol m$^{-2}$ s$^{-1}$ white light from a halogen lamp (HAL 100 W; Philips) installed on the confocal microscope. The light intensity of the halogen lamp was determined using a Lutron LX-120 light meter (Lutron, Taipei, Taiwan). Each step took 30 s of irradiance and images were immediately captured after each irradiance period. After 3 min with 30 s intervals of white light exposure, images were acquired from seedlings in darkness every 30 s for 4 min. The results of iNAP4 sensor were normalized with stromal iNAPc.

**Data analysis**. All data are presented as means with standard errors (mean ± SEM). The collected data were analysed for statistical significance using analysis of variance (ANOVA) with Tukey's HSD, paired t-tests, or unpaired t-tests at $P < 0.001$, $P < 0.01$, and $P < 0.05$ by SPSS (version 22).

**Reporting Summary**. Further information on research design is available in the Nature Research Reporting Summary linked to this article.

## Data availability

The authors declare that the main data supporting the findings of this study are available within the article and its Supplementary Information files. Image data have been uploaded to the repository Biostudies. The accession number is S-BIAD229. Extra data are available from the corresponding authors upon request. Source data are provided with this paper.

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

## Acknowledgements

This project was supported by the National Natural Science Foundation of China (31870212), the Seed Funding Program for Basic Research (201811159131), the Hong Kong Research Grants Council Area of Excellence Scheme (AoE/M-403/16), the Innovation and Technology Fund (Funding Support to State Key Laboratory of Agrobiotechnology) of the Hong Kong Special Administrative Region, China, the Swiss National Science Foundation SNSF-Grant 310030_185241 and the ETH Zürich. This research was also funded by a travel grant from the University Research Committee at the University of Hong Kong, which enabled SLL to conduct this research project at ETH Zurich. Any opinions, findings, conclusions or recommendations expressed in this publication do not reflect the views of the Government of the Hong Kong Special Administrative Region or the Innovation and Technology Commission. Data produced in this article were partially generated in collaboration with the Genetic Diversity Centre, ETH Zürich.

## Author contributions

B.L.L. conceptualized the research. B.L.L., S.L.L., S.F., and D.S. designed the study. S.F. performed RNA extraction, RT-qPCR, guard cell starch quantification of red light/ DCMU experiments, and gas exchange measurements. L.D. and S.L.L. optimised the GCC isolation protocol. JL carried out mesophyll chloroplast isolation experiments and GUS experiments. S.L.L. produced the transgenic iNAPs and SoNAR sensor lines and carried out all the other experiments. S.L.L. and S.F. analysed the data. B.L.L., S.L.L., S.F., and D.S. wrote the manuscript. All authors revised and approved the manuscript.

## Competing interests

The authors declare no competing interests.
