## [Peer Review File · Nature Communications]

Arabidopsis guard cell chloroplasts import cytosolic ATP for starch turnover and stomatal openingREVIEWER COMMENTS

Reviewer #1 (Remarks to the Author):

The paper from Lim, Flütsch and collaborators have investigated the function of guard cell chloroplasts (GCCs). Guard cells (GCs) have lower chloroplast/mitochondria ratio and very low level of chlorophyll, when compared to mesophyll cells (MCs). The plastidial photosynthetic activity was initially proposed to be absent in GCs. However, several recent works showed that GCCs are important for the regulation of stomatal opening, but the exact function of GCCs remain unclear. Thus, the work from Lim, Flütsch and collaborators is timely and address a long-debated question. By using different state of art techniques, the authors provide important information regarding ATP metabolism in GCs. However, the data produced do not support the conclusion that the major function of GCCs is to store starch and not to make photosynthesis. Contrary to this idea, the lack of starch synthesis under DCMU indicates a prominent role of photosynthesis in GCCs for starch synthesis.

I think the novelties and the main conclusions of the work should be:

- Mitochondria is the primary source of ATP for GCs
- GCCs import ATP through NTT, which is crucial for starch metabolism
- GCCs has considerable levels of both ATP and NADPH, despite low plastidial photosynthetic activity, as evidenced by the lack in increases in both ATP and NADPH upon illumination

Below the authors can find a thorough review of the work, with several suggestions to improve the text and the conclusions taken from the data.

Abstract: After reading the abstract, I don't see any result that confirm the affirmation found in the title of the work. Given the low level of chlorophyll found in GCs, no one is expecting high quantities of ATP and NADPH to be produced in this cell type.

Introduction: The introduction section has several erroneous concepts and misleading sentences. Furthermore, it didn't bring the hypothesis, aims and why this work is timely. The introduction is thus vague and contains out of date concepts. See below in detail.

Lines 19-20: "It remains controversial whether GCCs carry out photosynthesis." Not sure this assumption is correct. There are plenty of works showing that GCCs do photosynthesize. This idea is supported by the identification of rubisco in guard cell protoplasts by proteomics, by fluxomics studies and physiological measurements. The controversial idea is maybe based in the work from Outlaw (Outlaw et al., 1979).

Lines 33-34: "becomes turgid through the influx of potassium (K⁺), chloride (Cl⁻) and nitrate (NO₃⁻) ions, malate (Mal²⁻) and sugars" Is there any evidence showing that the influx of sugars makes GCs turgid? Perhaps the authors are basing in the role of sugars as osmolyte proposed by Zeiger and collaborators (Talbot & Zeiger, 1996), but this is solely based in correlation analysis. Alternatively, the authors are based in the results from Flütsch et al (Flütsch et al., 2020), in which reduced sugar transport reduced gs in Arabidopsis. None of these works have showed that sugars change the turgescence of GCs. Additionally, there are several works showing that exogenous sugar application leads to stomatal closure, meaning that sugars (and malate) can possibly decrease the turgescence of GCs.

Lines 43-44: "stomatal opening is energetically costly, consuming substantial amounts of cytosolic ATP 6" The authors cite a review here. It would be better to cite the original work that support such information.

Results:

Figure 1: I suggest comparing MCs and GCs in each time point.

If GCCs chloroplast function is not related to photosynthesis, why the NADPH level is higher at EoN and 3 min after illumination in GCs than MCs?

From EoN to 2 h, there is a decrease in both NADPH and ATP, suggesting a consumption of these components. Indeed, this is a critical period for stomatal opening. However, from 2h to 8 h, there is an increase in stromal ATP, this isn't an indication of photosynthetic activity?

Lines 86-87: "Given that the number of chloroplasts and the ratio of chloroplasts to mitochondria are much lower in GCs than in mesophyll cells 1,23" The reference 1 is a review. Please cite the original work that confirm it. Perhaps Willmer & Fricker has this information (Willmer & Fricker, 1996).

Lines 91-94: “By contrast, inhibitors of complex I (rotenone) and complex II (thenoyltrifluoroacetone, TTFA) of the mitochondrial transport chain partially decreased cytosolic ATP levels in mesophyll cells but had no significant effect on ATP levels in GCs (Supplementary Fig. 6a, b).” The trend is very similar among MCs and GCs. I think the best test here should be Dunnett, not Tukey, given that there is a clear control in the experiment (untreated cells).

Line 95: “These data suggest that mitochondria are the primary source of ATP for GCs” This is an outstanding result that solve a long debate. The authors should explore more this data in the discussion

Lines 103-104: “Given that plants expressing the TKTP-iNAP4 sensor in GCCs did not produce detectable amounts of NADPH in response to light” Sorry I’m lost here. This sentence is not supported by the Fig 1d. Did I miss something? Perhaps the authors want to state that not noticeable increases in NADPH was observed.

Lines 103-104: “These results further suggest that chloroplasts in both cell types contain substantial amounts of NADPH in the dark.” The amount of NADPH is much higher in guard cells. This should be discussed

Fig 2e. I’m wondering why the authors didn’t make kinetics of this analysis, such as in their previous work in PNAS

Lines 145-146: “DCMU inhibited starch accumulation in both WT and *stp1stp4* isolated GCs, indicating that GC photosynthesis at least partly contributed to starch synthesis (Fig. 3c).” The take home message here is clear: photosynthesis is crucial for starch synthesis, which is a paradox with the conclusions and the tittle of the work.

Fig. 3d. The protoplast of GCs is much lower than MCs. Thus, the expected is that the level of proteins will be lower than MCs. This data should be normalized to make this comparison a fair one. Perhaps the size of protoplasts or the level of chlorophyll can be used for the normalization.

Figures 3 g and h were not cited in the results section.

Discussion:

Lines 169-172: “The data presented here provide novel insights into how starch metabolism in GCs integrates and how GCCs obtain energy and carbon for starch metabolism when they do not generate sufficient ATP or NADPH during illumination nor fix substantial amount of CO₂ through the CBB pathway.” The authors haven’t measured CO₂ fixation. In general, the authors exacerbated and/or extrapolated their conclusions to unmeasured things.

Lines 176-178: “The ATP-dependent activation of the phototropin-H⁺-ATPase pathway triggers BL-dependent degradation of GC starch at the onset of illumination, which yields glucose, promoting rapid stomatal opening” I have two major concerns regarding this conclusion involving glucose. The major products exported from leaf plastidial starch degradation are maltose and glucose (Glc). It is unclear however which one (Glc or maltose) is the major product from starch breakdown in GCs. Evidence suggests that maltose and maltotriose are the main products of GC starch degradation (Talbot & Zeiger, 1993). Given that Flütsch et al. did not measure maltose, we cannot rule out the hypothesis that maltose is the main product of starch degradation released from GCCs. Furthermore, assuming that Glc is the major metabolite derived from starch degradation, one would expect Glc levels to increase after starch degradation, including in WT. However, in their experiment, Glc levels were unaltered from dark (D)-to-BL transition in WT plants. Given that a substantial part of the Glc pool is probably rapidly degraded to activate glycolysis under this condition, only a time series data would detect such increase in Glc content. Thus, the Glc data from single point measurements coupled to the lack of maltose quantification do not fully support the conclusion that Glc is the major metabolite derived from starch degradation.

In summary, the results obtained by Flütsch et al and the current work show compelling evidence for the role of guard cell starch metabolism during light stomatal responses. However, their attempts to answer (i) what is the major product of starch degradation and (ii) what is the fate of the C released from starch breakdown need to be interpreted with caution.

Lines 178-179: “The ability of GCCs to produce ATP and NADPH is marginal (Fig. 1b, d)” I don’t see tremendous differences between MCs and GCs in both ATP and NADPH levels. The NADPH seems even higher in GCs than MCs. The authors should rephrase this sentence or provide statistical comparison between the two cell types.

Lines 188-189: “Hence, the main function of GCCs is not photosynthesis or CO₂ fixation; rather, it is to serve as a reservoir of starch, which is essential for regulating stomatal opening” What are the evidence for this conclusion? Is not clear to me. I think the work provide evidence for the opposite conclusion: that photosynthesis is crucial for starch synthesis in illuminated GCs. This idea is based in the fact that DCMU treatment abolished starch synthesis in GCs.

Another major concern: GCs have several characteristics of sink cells and C₄/CAM cells. Thus, in order to fully understand the function of GCCs, the authors should make a wider comparison, using source and sink MCs of the same plant and possibly MCs from other species.

References cited here:

Flütsch S, Nigro A, Conci F, Fajkis J, Thalmann M, Trtílek M, Panzarová K, Santelia D. 2020. Glucose uptake to guard cells via STP transporters provides carbon sources for stomatal opening and plant growth. *EMBO reports* in press: 1–13.

Outlaw WH, Manchester J, Dicamelli CA, Randall DD, Rapp B, Veith GM. 1979. Photosynthetic carbon reduction pathway is absent in chloroplasts of *Vicia faba* guard cells. *Proceedings of the National Academy of Sciences of the United States of America* 76: 6371–5.

Talbott LD, Zeiger E. 1993. Sugar and organic acid accumulation in guard cells of *Vicia faba* in response to red and blue light. *Plant Physiology* 102: 1163–1169.

Talbott LD, Zeiger E. 1996. Central Roles for Potassium and Sucrose in Guard-Cell Osmoregulation. *Plant physiology* 111: 1051–1057.

Willmer C, Fricker M. 1996. *Stomata*. London: Chapman & Hall.

Reviewer #2 (Remarks to the Author):

The MS by Lim et al explores the function of guard cell chloroplasts in *Arabidopsis*. Using a range of fluorescence protein sensors the authors monitored ATP and NADPH production in guard cells and compared the findings to mesophyll cells and concluded that photosynthetic electron transport is not responsible for the production of ATP required for stomatal opening to blue light. Using Nucleotide transporter mutants, they suggest that ATP is imported into guard cell chloroplasts and this is required for starch biosynthesis. From these findings the authors conclude that guard cell chloroplasts serve as a starch reservoir rather than carrying out photosynthesis.

This is an interesting MS as the role of guard chloroplasts has been debated for many years (although work by others in this area has not been included in the MS – see comment below). The authors have carried out a substantial amount of work and used a range of different approaches to explore guard cell chloroplast metabolism. Below I have outlined some comments to help improve the MS.

My main concern with the MS regards the use of only fluorescence probes to “quantify” the amounts of substances. These ratiometric probes can be extremely useful for cellular localization and provide some indicative measure of substances - however they are also subject to various limitations, with different re-absorption properties depending on the wavelength, pH and redox influences as well as detection. Furthermore, confocal microscopy can easily damage the material and probes due to photo bleaching episodes. Putting these caveats aside I am not entirely convinced by the differences between guard and mesophyll cell signals suggested by the authors. It is extremely difficult to see the differences between the mesophyll and the guard cells in Suppl Fig.1. This is supported by the fact that the absolute levels outlined in Fig. 1 and suppl Fig.2 are in very similar ranges and that only very small differences between the two cell types were observed when illuminated. This could be because the probes/sensors are saturated –did the authors test for this? These data are not convincing and it would be nice to see these supported by “actual” measurements rather than proxy measures.

How do the authors explain such high levels of NADPH in the dark?

The authors should explain exactly how the dyes work and include the known pitfalls with these radiometric dyes.

Why was 3-minute illumination chosen for these measurements?

What does high and low represent in the false colour images? This information should be included in the colour scale bar and are the scale bars the same for all images?

The authors present data illustrating that GC contain much lower amounts of CBB cycle enzymes than mesophyll. This is not new and has been demonstrated and discussed previously. GCs are 20x smaller than mesophyll was this taken into account?

Can the authors provide an explanation for the presence of Calvin cycle enzymes in the guard cells if no Calvin cycle activity is taking place? Fig. 3d – the authors show different amounts of starch produced in the ntt mutants compared with wild type – although the ntt1 and ntt2 mutants did not accumulate starch the stomata were still functional with only small differences in aperture observed, and stomata remained open for 8 h. What spectrum of light was used in these experiments?

The introduction does not fully cover the literature on guard cells photosynthesis – there are several reviews in this area that have not been included. The amount of information in the blue light pathway could be reduced to a few sentences and focus only on the fact that energy (ATP) is a key component of this pathway. Additionally the authors need to include more context to highlight why the information from this study is important.

The mutants need to be introduced fully to the reader. Difference between NTT1 and NTT2 etc.

Can the authors provide further explanations for the effect of complex I and II inhibitors relative to oligomycin on the effects of mitochondrial ATP production?

The authors demonstrate that guard cell electron transport does contribute to some production of starch and this should be included and discussed further in the discussion section.

All of the figure legends need to state which light spectra were used – blue, red or white light.

The discussion is a little repetitive of the results in places. It would be good to see further discussion on the context. Also the authors focus on BL stomatal responses – however what about the red light driven pathway – could guard cell metabolism be different under different conditions? It should be acknowledged that this work focuses specifically on the BL pathway.

Letter is missing from Fig. 3d.

The symbols in the graphs of Fig 3 are extremely difficult to distinguish between controls and treatments. I would recommend using filled and open symbols rather than various greys.

Suppl Fig. 7 – the untreated GC at 8 h in the dark appear at a similar YPF level as guard cells at time zero EoN guard cells – can the authors discuss these finding in greater detail?

Line 44 – substantial amounts – what is substantial?

The following new figures were added to the revised manuscript:

Fig. 2a-f, Fig. 3c-f, Fig. 5d-e, Supplementary Fig 7a-d.

**REVIEWER COMMENTS**

Reviewer #1 (Remarks to the Author):

The paper from Lim, Flütsch and collaborators have investigated the function of guard cell chloroplasts
(GCCs). Guard cells (GCs) have lower chloroplast/mitochondria ratio and very low level of chlorophyll,
when compared to mesophyll cells (MCs). The plastidial photosynthetic activity was initially proposed to
be absent in GCs. However, several recent works showed that GCCs are important for the regulation of
stomatal opening, but the exact function of GCCs remain unclear. Thus, the work from Lim, Flütsch and
collaborators is timely and address a long-debated question. By using different state of art techniques, the
authors provide important information regarding ATP metabolism in GCs. However, the data produced
do not support the conclusion that the major function of GCCs is to store starch and not to make
photosynthesis. Contrary to this idea, the lack of starch synthesis under DCMU indicates a prominent role
of photosynthesis in GCCs for starch synthesis.

> We agree with the reviewer's comment. We revised the manuscript title to "Arabidopsis guard cell
chloroplasts import cytosolic ATP for starch turnover and stomatal opening" and revised the text
accordingly. Specifically, we now have a new paragraph in the result section entitled "DCMU inhibits
starch synthesis in isolated GCs" (Line 211).

I think the novelties and the main conclusions of the work should be:

- - Mitochondria is the primary source of ATP for GCs
- - GCCs import ATP through NTT, which is crucial for starch metabolism
- - GCCs has considerable levels of both ATP and NADPH, despite low plastidial photosynthetic activity,
as evidenced by the lack in increases in both ATP and NADPH upon illumination.

> The reviewer correctly points out the main conclusions of our work. In the revised discussion, we
discuss them in much greater details.

Below the authors can find a thorough review of the work, with several suggestions to improve the text
and the conclusions taken from the data.

Abstract: After reading the abstract, I don't see any result that confirm the affirmation found in the title of
the work. Given the low level of chlorophyll found in GCs, no one is expecting high quantities of ATP
and NADPH to be produced in this cell type.

> We agree with the reviewer's comment. We revised the manuscript title to "Arabidopsis guard cell
chloroplasts import cytosolic ATP for starch turnover and stomatal opening" and revised the abstract and
the overall text accordingly.

Introduction: The introduction section has several erroneous concepts and misleading sentences.

Furthermore, it didn't bring the hypothesis, aims and why this work is timely. The introduction is thus
vague and contains out of date concepts. See below in detail.

> We appreciate the reviewer's comment. We carefully revised the introduction to include the reviewer's
suggestions and to put the work into its wider up-to-date context, including a clear statement of its
rationale.

Lines 19-20: “It remains controversial whether GCCs carry out photosynthesis.” Not sure this assumption
is correct. There are plenty of works showing that GCCs do photosynthesize. This idea is supported by
the identification of rubisco in guard cell protoplasts by proteomics, by fluxomics studies and
physiological measurements. The controversial idea is maybe based in the work from Outlaw (Outlaw et
al., 1979).

> The reviewer is correct. There is literature supporting the occurrence of GC photosynthesis and our data
ultimately point towards the idea that at least in part GCCs carry photosynthesis, although at levels that
may not be enough to fully power chloroplast metabolism (both in terms of ATP, NADPH generation and
provision of carbon skeletons). In the revised version of the manuscript, this sentence is therefore
removed to avoid confusion. And we now discuss in great details the contribution of GC photosynthesis
to starch metabolism and in general to stomatal movements.

Lines 33-34: “becomes turgid through the influx of potassium (K⁺), chloride (Cl⁻) and nitrate (NO₃⁻)
ions, malate (Mal²⁻) and sugars” Is there any evidence showing that the influx of sugars makes GCs
turgid? Perhaps the authors are basing in the role of sugars as osmolyte proposed by Zeiger and
collaborators (Talbot & Zeiger, 1996), but this is solely based in correlation analysis. Alternatively, the
authors are based in the results from Flütsch et al (Flütsch et al., 2020), in which reduced sugar transport
reduced gs in Arabidopsis. None of these works have showed that sugars change the turgescence of GCs.
Additionally, there are several works showing that exogenous sugar application leads to stomatal closure,
meaning that sugars (and malate) can possibly decrease the turgescence of GCs.

> The reviewer makes here an excellent point. None of the current literature directly demonstrates that
sugars produced by guard cells or imported from the mesophyll act directly as osmolytes to change guard
cell turgor. The sentence has been therefore rephrased by using the wording “inorganic” and “organic”
ions (line 35). We additionally point out in the revised introduction at line 44 that “the precise function of
sugars within the guard cells is not fully understood”.

Lines 43-44: “stomatal opening is energetically costly, consuming substantial amounts of cytosolic ATP
6” The authors cite a review here. It would be better to cite the original work that support such
information.

> Thank you. We have cited original research papers describing the work supporting this statement.

- 1. Ref. 14: Tominaga, et al. “Guard-cell chloroplasts provide ATP required for H⁺ pumping in the
plasma membrane and stomatal opening” *Plant & Cell Physiology* (2001), 42:795-802.
- 2. Ref. 15. Wang, et al. “Lacking chloroplasts in guard cells of crumpled leaf attenuates stomatal
opening: both guard cell chloroplasts and mesophyll contribute to guard cell ATP levels” *Plant,
Cell and Environment* (2014), 37:2201-2210.

Results:

Figure 1: I suggest comparing MCs and GCs in each time point.

> As the ATP and NADPH concentrations are affected by their production/consumption and the size of
the subcellular compartments, a direct comparison between MC and GC is not advisable. To address the
reviewer’s suggestion, we have performed an additional experiment to compare the real-time dynamic
changes of ATP and NADPH in the same subcellular compartment (the chloroplast stroma) of different
cell types (i.e., root, guard cells, mesophyll cells) upon various light treatments. We have added these data
in the revised version as Fig. 2.

If GCCs chloroplast function is not related to photosynthesis, why the NADPH level is higher at EoN and
3 min after illumination in GCs than MCs?

> The reviewer poses an excellent question. It is known that the relative level of stromal NADPH is
affected by its production and consumption. The NADPH synthesis pathways include the oxidative
pentose phosphate pathway (OPPP) and the light reactions of photosynthesis. The NADPH consumption
pathways include nitrite reduction and the Calvin Benson Cycle (CBB) pathway. To address the
contribution of the OPPP pathway to NADPH synthesis in GCs, we used 6-aminonicotinamide (6-AN), a
phosphogluconate dehydrogenase inhibitor, to block the OPPP. Our new data (Fig. 3c-d) showed that 6-
AN treatment can reduce NADPH levels in both GCs and MCs, suggesting the high NADPH level at the
EoN is due to the OPPP activity which normally occurs in plastid stroma in the dark. Given the GCs have
very limited photosynthetic activity, we suggest that the higher levels of NADPH at the EoN in GCs
compared to MCs are likely a direct result of the OPPP activity, which may represent the main source of
NADPH for guard cells.

We have added a dedicated paragraph in the results section to discuss this important set of data. The
reviewer will find this starting at line 105 with the title “GCCs have higher levels of NADPH in the dark
than MCCs”.

From EoN to 2 h, there is a decrease in both NADPH and ATP, suggesting a consumption of these
components. Indeed, this is a critical period for stomatal opening. However, from 2h to 8 h, there is an
increase in stromal ATP, this isn't an indication of photosynthetic activity?

> The levels of stromal ATP and NADPH at any given time are the balance of various production and
consumption pathways. Our data showed that GC but not mature mesophyll cells express NTTs (Fig. 4a,
Fig. 4b and Supplementary Fig. 4a-b) and therefore ATP can enter GCCs but not mature mesophyll
chloroplasts (Fig. 4c-d). Since photosynthetic activity of GC chloroplast is low, as supported by the
insignificant changes of stromal pH upon illumination (Fig. 1f), it is likely that the major source of
stromal ATP in GCCs is the entry of cytosolic ATP, which were similar between EoN and 8h of light
(Fig. 5a). Assuming the rate of ATP import into guard cell chloroplasts is constant, the lower stromal
ATP at T = 2 could be due to its higher consumption, for example, by starch synthesis. Our data showed
that GCCs and mesophyll chloroplasts have different sources of ATP. While mesophyll chloroplasts
mainly rely on photosynthesis, GCCs mainly rely on the import of cytosolic ATP.

Lines 86-87: “Given that the number of chloroplasts and the ratio of chloroplasts to mitochondria are
much lower in GCs than in mesophyll cells 1,23” The reference 1 is a review. Please cite the original
work that confirm it. Perhaps Willmer & Fricker has this information (Willmer & Fricker, 1996).

> Thank you. We have cited the original work (Ref. 23) (Allaway and Setterfield, 1972).

Lines 91-94: “By contrast, inhibitors of complex I (rotenone) and complex II (thenoyltrifluoroacetone,
TTFA) of the mitochondrial transport chain partially decreased cytosolic ATP levels in mesophyll cells
but had no significant effect on ATP levels in GCs (Supplementary Fig. 6a, b).” The trend is very similar
among MCs and GCs. I think the best test here should be Dunnett, not Tukey, given that there is a clear
control in the experiment (untreated cells).

> Thank you. We tried to use the one-way ANOVA with Dunnett's test and generated a new figure.

However, after adding a combinatory treatment of both rotenone and TTFA to Fig. 3e,f, the trends
 became similar among MCs and GCs when one-way ANOVA and Tukey's HSD test ($P < 0.05$) were
 used. Hence, we updated the manuscript with the figure below and revised the text accordingly.

95: "These data suggest that mitochondria are the primary source of ATP for GCs" This is an outstanding
 result that solve a long debate. The authors should explore more this data in the discussion
 > Thank you. To strengthen our conclusions, we performed an additional experiment in which the
 treatment of rotenone + TTFA were combined (Fig. 3e-f). These new results clearly showed that
 mitochondria are indeed the primary source of cytosolic ATP for GCs. We have revised the discussion to
 better highlight this conclusion, including discussion of the currently available literature.

Lines 103-104: "Given that plants expressing the TKTP-iNAP4 sensor in GCCs did not produce
 detectable amounts of NADPH in response to light" Sorry I'm lost here. This sentence is not supported by
 the Fig 1d. Did I miss something? Perhaps the authors want to state that not noticeable increases in
 NADPH was observed.

> We apologise with the reviewer if the wording created some confusion. At lines 110-112, we amended
 the sentence to "The levels of NADPH at the EoN in GCs were even higher than those observed in MCs,
 although no noticeable increases in stromal NADPH were observed in GCCs upon illumination (Fig. 1d
 and Fig. 2d),".

Lines 103-104: "These results further suggest that chloroplasts in both cell types contain substantial
 amounts of NADPH in the dark." The amount of NADPH is much higher in guard cells. This should be
 discussed

> We thank you both reviewers for highlighting this interesting difference. We have revised the
 manuscript, adding a dedicated paragraph in the result section to discuss these data in more detail (Lines
 105-123).

Fig 2e. I'm wondering why the authors didn't make kinetics of this analysis, such as in their previous
 work in PNAS

> We thank you the reviewer for this suggestion. Accordingly, we performed an additional experiment to

measure the kinetics of ATP and NADPH production. The results are presented in the revised manuscript
in Fig. 2a-f. Our kinetic analysis showed that illumination stimulated ATP and NADPH production and
alkalization in mesophyll chloroplasts but not in GCCs and root.

Lines 145-146: “DCMU inhibited starch accumulation in both WT and *stp1stp4* isolated GCs, indicating
that GC photosynthesis at least partly contributed to starch synthesis (Fig. 3c).” The take home message
here is clear: photosynthesis is crucial for starch synthesis, which is a paradox with the conclusions and
the title of the work.

> We agree that GC photosynthesis at least partly contributed to starch synthesis. Following the
reviewer’s suggestion, we revised the manuscript title to “*Arabidopsis* guard cell chloroplasts import
cytosolic ATP for starch turnover and stomatal opening” and revised the text accordingly, with a new
paragraph discussing the DCMU data in greater details (Lines 239-247).

Fig. 3d. The protoplast of GCs is much lower than MCs. Thus, the expected is that the level of proteins
will be lower than MCs. This data should be normalized to make this comparison a fair one. Perhaps the
size of protoplasts or the level of chlorophyll can be used for the normalization.

> The reviewer makes a valuable comment. To take this into consideration, please see in Fig. 5i that the
protein concentrations of GCs and MCs were quantified and equalized before loading. The Western Blot
data showed that the amounts of actin, a housekeeping protein, were indeed comparable between GCs and
MCs, while other key proteins showed different amounts between the two cell types. Since GCCs and
mesophyll chloroplasts greatly differ in size (Supplementary Fig. 5b), protoplast size or the level of
chlorophyll may vary a lot and it is better not to use them for normalization of protein amounts.

Figures 3 g and h were not cited in the results section.

> We apologise for this mistake. In the revised manuscript, we discuss Fig. 3g and 3h (now
Supplementary Fig. 3c-d) in line 150.

Discussion:

Lines 169-172: “The data presented here provide novel insights into how starch metabolism in GCs
integrates and how GCCs obtain energy and carbon for starch metabolism when they do not generate
sufficient ATP or NADPH during illumination nor fix substantial amount of CO₂ through the CBB
pathway.” The authors haven’t measured CO₂ fixation. In general, the authors exacerbated and/or
extrapolated their conclusions to unmeasured things.

> The reviewer is right in pointing out that we have not directly measured CO₂ fixation in GCs. However,
other labs have previously used O₂ evolution in guard cell- and mesophyll protoplasts (Shimazaki and
Okayama, 1990), or incorporation of carbon isotopes into metabolic intermediates (Daloso et al., 2015;
Shimazaki and Okayama, 1990) as a proxy for photosynthetic activity. In all cases, these authors have
demonstrated a modest but detectable photosynthetic activity in guard cells. Our Western Blot results,
showing that GCs express CBB enzymes at a much lower levels compared to mesophyll cells, similarly
suggest that only a modest CO₂ fixation activity occurs in GCCs compared to the mesophyll cell
chloroplasts. This conclusion is further supported by the DCMU data presented in Fig. 5g-h. Thus, the
above-mentioned sentence is based on both data from the literature and our own experiments.

Lines 176-178: “The ATP-dependent activation of the phototropin-H⁺-ATPase pathway triggers BL-

dependent degradation of GC starch at the onset of illumination, which yields glucose, promoting rapid
stomatal opening” I have two major concerns regarding this conclusion involving glucose. The major
products exported from leaf plastidial starch degradation are maltose and glucose (Glc). It is unclear
however which one (Glc or maltose) is the major product from starch breakdown in GCs. Evidence
suggests that maltose and maltotriose are the main products of GC starch degradation (Talbot & Zeiger,
1993). Given that Flütsch et al. did not measure maltose, we cannot rule out the hypothesis that maltose is
the main product of starch degradation released from GCCs. Furthermore, assuming that Glc is the major
metabolite derived from starch degradation, one would expect Glc levels to increase after starch
degradation, including in WT. However, in their experiment, Glc levels were unaltered from dark (D)-to-
BL transition in WT plants. Given that a substantial part of the Glc pool is probably rapidly degraded to
activate glycolysis under this condition, only a time series data would detect such increase in Glc content.
Thus, the Glc data from single point measurements coupled to the lack of maltose quantification do not
fully support the conclusion that Glc is the major metabolite derived from starch degradation.

In summary, the results obtained by Flütsch et al and the current work show compelling evidence for the
role of guard cell starch metabolism during light stomatal responses. However, their attempts to answer
(i) what is the major product of starch degradation and (ii) what is the fate of the C released from starch
breakdown need to be interpreted with caution.

> We thank you the reviewer for this thoughtful comment. We fully agree that single time point
measurements do not reveal metabolic dynamics and their kinetics. In guard cells, measure of metabolic
kinetics is a challenging task and would only be possible by using isotope labelling. As such, we cannot
exclude that in Flütsch et al, (Flütsch et al., 2020), a transient peak in malate and/or glucose accumulation
was missed due to our experimental setup. Simultaneous synthesis/use of malate in guard cells and rapid
consumption of glucose through glycolysis makes it difficult to detect fine changes in the amount of these
metabolites. However, the fact that there were no differences in malate content between the wild type and
*amy3 bam1* mutant led us to conclude that malate is not the major starch-derived metabolite in
*Arabidopsis* guard cells during BL-induced stomatal opening. As for maltose: this is usually measured by
HPLC, as there is no sensitive assay that can reliably measure maltose enzymatically. In any case, even if
maltose was exported from the guard cell chloroplast, this would be rapidly converted into glucose
through a series of cytosolic reactions involving for example enzymes such as the disproportionating
enzyme DPE2 and the alpha-glucan phosphorylase PHS2. The questions (i) what is the major product of
starch degradation and (ii) what is the fate of the C released from starch breakdown are still open and
research in our lab is currently attempting to provide an answer.

Lines 178-179: “The ability of GCCs to produce ATP and NADPH is marginal (Fig. 1b, d)” I don’t see
tremendous differences between MCs and GCs in both ATP and NADPH levels. The NADPH seems
even higher in GCs than MCs. The authors should rephrase this sentence or provide statistical comparison
between the two cell types.

> We apologise with the reviewer if this sentence generated confusion. Here, we mean that the increase in
FRET ratio upon illumination in GCCs is not significant, suggesting that the ability of GCCs to produce
ATP and NADPH in response to light is marginal. We have rephrased the sentence to make this clear.

Lines 188-189: “Hence, the main function of GCCs is not photosynthesis or CO₂ fixation; rather, it is to serve as a
reservoir of starch, which is essential for regulating stomatal opening” What are the evidence for this conclusion? Is
not clear to me. I think the work provide evidence for the opposite conclusion: that photosynthesis is crucial for
starch synthesis in illuminated GCs. This idea is based in the fact that DCMU treatment abolished starch synthesis in
GCs.

> We agree with the reviewer’s comment. We remove this sentence and in the revised manuscript we

have dedicated sections in both results and discussion to highlight that GC photosynthesis is important for
starch synthesis.

Another major concern: GCs have several characteristics of sink cells and C4/CAM cells. Thus, in order
to fully understand the function of GCCs, the authors should make a wider comparison, using source and
sink MCs of the same plant and possibly MCs from other species.

> The reviewer correctly points out that GCs have several features of sink cells. The suggestion to
compare FRET changes in source and sink MCs within the same rosette is excellent. Indeed, a similar
experiment was done previously in Arabidopsis using $^{14}\text{CO}_2$ pulse-chase labelling. It was shown that a
small developing leaf which acts as sink MCs (i.e. leaf no. 13 of a 3-week-old plant) is still able to
photosynthesize when fed with $^{14}\text{CO}_2$. However, it does not export carbon to the neighbouring cells nor to
other sink tissues (Kölling et al., 2013; Kölling et al., 2015). Based on this evidence, we assume that GCs
will behave similarly in source and sink MCs.

Arguably, a better comparison would be between GCs and other sink tissues, such as root cells. Thus,
prompted by the reviewer suggestion to expand our analysis, we have performed an additional
experiment, in which we measured real-time dynamic changes of ATP and NADPH in the same
subcellular compartment (the plastid stroma) of different cell types (i.e. root, guard cells, mesophyll cells)
upon various light treatments (Fig. 2). The results showed that GCs behave very similarly to root cells,
confirming their characteristics of sink tissues. However, GCs showed overall higher amounts of ATP
and NADPH, and a more alkaline stroma compared to roots. It seems likely that even if GCs do not
produce significant amounts of ATP and NADPH through photosynthesis, their metabolism is adapted to
supply the demand of energy for stomatal movements.

Reviewer #2 (Remarks to the Author):

The MS by Lim et al explores the function of guard cell chloroplasts in Arabidopsis. Using a range of
fluorescence protein sensors the authors monitored ATP and NADPH production in guard cells and
compared the findings to mesophyll cells and concluded that photosynthetic electron transport is not
responsible for the production of ATP required for stomatal opening to blue light. Using Nucleotide
transporter mutants, they suggest that ATP is imported into guard cell chloroplasts and this is required for
starch biosynthesis. From these findings the authors conclude that guard cell chloroplasts serve as a starch
reservoir rather than carrying out photosynthesis.

This is an interesting MS as the role of guard chloroplasts has been debated for many years (although
work by others in this area has not been included in the MS – see comment below). The authors have
carried out a substantial amount of work and used a range of different approaches to explore guard cell
chloroplast metabolism. Below I have outlined some comments to help improve the MS.

My main concern with the MS regards the use of only fluorescence probes to “quantify” the amounts of
substances. These ratiometric probes can be extremely useful for cellular localization and provide some
indicative measure of substances - however they are also subject to various limitations, with different re-
absorption properties depending on the wavelength, pH and redox influences as well as detection.

Furthermore, confocal microscopy can easily damage the material and probes due to photo bleaching

episodes. Putting these caveats aside I am not entirely convinced by the differences between guard and
mesophyll cell signals suggested by the authors. It is extremely difficult to see the differences between the
mesophyll and the guard cells in Suppl Fig.1. This is supported by the fact that the absolute levels
outlined in Fig. 1 and suppl Fig.2 are in very similar ranges and that only very small differences between
the two cell types were observed when illuminated.

This could be because the probes/sensors are saturated –did the authors test for this? These data are not
convincing and it would be nice to see these supported by “actual” measurements rather than proxy
measures.

> The reviewer makes a very good point. It would be in theory great to confirm our FRET analyses with
biochemical measurements. However, absolute metabolite quantification in guard cells is still technically
very challenging, especially for molecules such as NADPH and ATP, which have a very short half-life.
Previously, ATP in guard cells has been measured by using a bioluminescence assay kit in epidermal
peels of Arabidopsis luciferase transgenics plants and expressed as relative luminescence intensity/cell
(Wang et al., 2014). However, no subcellular information could be determined by this method.

We decided to use the sensors. Despite the sensors are not able to provide quantitative information about
absolute metabolite amounts, they are very useful in detecting relative changes *in planta* (i.e. dark vs
illuminated), which was the aim of our study. Furthermore, the sensors expressed in different subcellular
compartments give the advantage to track organelle-specific changes in ATP amounts. Overall, we
believe that considering current technical limitations, FRET sensors are a great tool to study ATP and
NADPH metabolism in guard cells.

The *in planta* performance of these sensors have been carefully examined in our previous studies (Lim et
al., 2020; Voon et al., 2018). In any case, to address the reviewer’s comment and further show the
responsiveness of these sensors *in planta*, we carried out an additional experiment in which we have
examined their kinetic performance upon illumination. As shown in Fig. 2a-b, the stromal ATP
concentration was not saturated in the dark: before illumination, the ratios were comparable in both types
of chloroplasts, while illumination only increased the ATP concentration in mesophyll chloroplasts but
not GCCs. Similar observation could be seen for the pH sensor (Fig. 2e-f). NADPH was also not
saturated (Fig. 2c-d). In GCCs, the stromal NADPH concentrations were lower at 2h and 8 h into the day
compared to the EoN (Fig. 1d). However, no detectable increase in stromal NADPH were observed upon
illumination at these two time points. In the revised manuscript, we now clearly state that our sensors
were operating within their respective optimal range of detection.

How do the authors explain such high levels of NADPH in the dark?

The authors should explain exactly how the dyes work and include the known pitfalls with these
radiometric dyes.

> Please, see our answer to reviewer 1 starting at line 92 of this file. In M&M, we cite the original papers
describing each sensor, where the reader will find precise information about how each sensor works and
their advantages/disadvantages.

Why was 3-minute illumination chosen for these measurements?

> Photosynthesis responses to light are known to be quick. For example, upon changes in photosynthetic
photon flux density during sun/shade flecks caused by passing clouds or overlapping leaves in a canopy,
photosynthesis adapts quickly by reaching a new steady state within several tens of seconds to minutes
(Barradas and Jones, 1996). Thus, 3 min should be largely sufficient to observe changes in photosynthesis
activity both in guard cell and mesophyll cells. The same experimental conditions were indeed used in our

previous studies (Lim *et al.*, 2020; Voon *et al.*, 2018).

What does high and low represent in the false colour images? This information should be included in the
colour scale bar and are the scale bars the same for all images?

> We amended the colour bars. We now present them with a more precise description (eg, low/ high
MgATP²⁻, low/high NADPH level, and acid/ alkaline) and included the information of each scale bar in
the figure legend.

The authors present data illustrating that GC contain much lower amounts of CBB cycle enzymes than
mesophyll. This is not new and has been demonstrated and discussed previously. GCs are 20x smaller
than mesophyll was this taken into account?

> To the best of our knowledge, previous reports focused on other plant species. No western blot data on
CBB enzymes are available for Arabidopsis guard cells. While we recognize that our data confirm
findings from other species (which in a way is really nice!), we believe that they play a key role in our
manuscript as they match with the results of the sensor experiments and DCMU treatment, and so provide
further support to our conclusions. Given the differ in size between GCs and MCs, we normalized the
protein loading to the housekeeping protein actin and compared a number of cytosolic enzymes and
stromal enzymes.

Can the authors provide an explanation for the presence of Calvin cycle enzymes in the guard cells if no
Calvin cycle activity is taking place?

> We did not say there was no CBB activity in GC.

Fig. 3d – the authors show different amounts of starch produced in the *ntt* mutants compared with wild
type – although the *ntt1* and *ntt2* mutants did not accumulate starch the stomata were still functional with
only small differences in aperture observed, and stomata remained open for 8 h. What spectrum of light
was used in these experiments?

> The stomatal aperture measurements were performed under white light at 150 μ E. However, as the
reviewer will see in our revised manuscript, the stomatal aperture results have been replaced with gas
exchange experiments using LI-COR 6400 (Fig. 5d-e, Supplementary Fig. 7). Measuring stomatal
aperture under the microscope is not a very precise method. Instead, the use of Licor allows fine stomatal
conductance (g_s) measurements, providing important information about the kinetics of stomatal
movements (both rapidity of opening and closing, and amplitude of opening response). The new g_s data
revealed that *ntt1* mutant has dramatic reductions in g_s responses upon transition from dark to light and
light to dark compared to wild type, while *ntt2* mutant has a milder phenotype. The g_s phenotype of *ntt*
mutants reflects their relative expression levels in guard cells, as reported in Fig. 4a (*NTT1* is more highly
expressed in guard cells compared to *NTT2*). The guard cell starch phenotype is also interesting, as you
see that *ntt2* mutant shows some starch accumulation after 8 h of light, while *ntt1* mutant remains
essentially devoid of starch throughout the day.

The introduction does not fully cover the literature on guard cells photosynthesis – there are several
reviews in this area that have not been included. The amount of information in the blue light pathway
could be reduced to a few sentences and focus only on the fact that energy (ATP) is a key component of
this pathway. Additionally the authors need to include more context to highlight why the information
from this study is important.

> The introduction has been substantially revised to take the reviewer's comments and suggestions into
consideration. Thank you.

The mutants need to be introduced fully to the reader. Difference between NTT1 and NTT2 etc.

> We added detailed information about NTT1 and NTT2 at lines 159-165 of the revised manuscript

Can the authors provide further explanations for the effect of complex I and II inhibitors relative to
oligomycin on the effects of mitochondrial ATP production?

> Rotenone and TTFA only inhibits complex I and complex II, respectively. Both complexes partially
contribute reducing equivalents to the mETC. Hence, unlike oligomycin, which completely inhibits
mitochondrial ATP production, the use of either rotenone or TTFA could only partially reduce the ATP
concentration. To fully address the reviewer comment, we performed additional FRET experiment using
both rotenone and TTFA inhibitors, as shown in Fig. 3e-f. The combined use of rotenone and TTFA has
greater impact on ATP production compared to the single inhibitor treatments, particularly in guard cells.

The authors demonstrate that guard cell electron transport does contribute to some production of starch
and this should be included and discussed further in the discussion section.

> The discussion has been extensively revised to take this and other reviewer's comments into
consideration.

All of the figure legends need to state which light spectra were used – blue, red or white light.

> Light spectra information is now inserted in each figure legend.

The discussion is a little repetitive of the results in places. It would be good to see further discussion on
the context. Also the authors focus on BL stomatal responses – however what about the red light driven
pathway – could guard cell metabolism be different under different conditions? It should be
acknowledged that this work focuses specifically on the BL pathway.

> The discussion has been extensively revised to take this and other reviewer's comments into
consideration.

Letter is missing from Fig. 3d.

> This has been amended.

The symbols in the graphs of Fig 3 are extremely difficult to distinguish between controls and treatments.
I would recommend using filled and open symbols rather than various greys.

> We replotted the graphs of Fig. 3 (now Fig. 5h), using different colours instead of shades of grey.
Thank you for the suggestion.

Suppl Fig. 7 – the untreated GC at 8 h in the dark appear at a similar YPF level as guard cells at time zero
EoN guard cells – can the authors discuss these finding in greater detail?

> The sensor ratio of the untreated GC at 8h was lower than the untreated GC at EoN (now
Supplementary Fig. 3). This likely indicates that after prolonged illumination the GC cytosol became
more acidic. However, this is our interpretation and we feel we cannot provide a simple explanation for
this observation as cytosolic pH is subject to a variety of controlling factors.

Line 44 – substantial amounts – what is substantial?

> We removed the word substantial and instead use “high”. It is difficult to estimate exactly how much

ATP is needed, as this is also species-specific. As we mentioned above, absolute quantification of
metabolites such as ATP or NADPH is very difficult, particularly in the tiny guard cells.

**References:**

Allaway, W.G., and Setterfield, G. (1972). Ultrastructural observations on guard cells of *Vicia Faba* and
*Allium Porrum*. *Can J Botany* 50, 1405-1413.

Barradas, V.L., and Jones, H.G. (1996). Responses of CO₂ assimilation to changes in irradiance:
laboratory and field data and a model for beans (*Phaseolus vulgaris* L.). *Journal of Experimental Botany*
47, 639-645. 10.1093/jxb/47.5.639.

Daloso, D.M., Antunes, W.C., Pinheiro, D.P., Waquim, J.P., Araujo, W.L., Loureiro, M.E., Fernie, A.R., and
Williams, T.C. (2015). Tobacco guard cells fix CO₂ by both Rubisco and PEPcase while sucrose acts as a
substrate during light-induced stomatal opening. *Plant, cell & environment* 38, 2353-2371.

10.1111/pce.12555.

Flütsch, S., Wang, Y., Takemiya, A., Violet-Chabrand, S.R., Klejchova, M., Nigro, A., Hills, A., Lawson, T.,
Blatt, M.R., and Santelia, D. (2020). Guard cell starch degradation yields glucose for rapid stomatal
opening in *Arabidopsis*. *Plant Cell* 32, 2325-2344. 10.1105/tpc.18.00802.

Kölling, K., Müller, A., Flütsch, P., and Zeeman, S.C. (2013). A device for single leaf labelling with CO₂
isotopes to study carbon allocation and partitioning in *Arabidopsis thaliana*. *Plant Methods* 9, 45.

10.1186/1746-4811-9-45.

Kölling, K., Thalmann, M., Müller, A., Jenny, C., and Zeeman, S.C. (2015). Carbon partitioning in
*Arabidopsis thaliana* is a dynamic process controlled by the plants metabolic status and its circadian
clock. *Plant, cell & environment* 38, 1965-1979. <https://doi.org/10.1111/pce.12512>.

Lim, S.L., Voon, C.P., Guan, X., Yang, Y., Gardestrom, P., and Lim, B.L. (2020). *In planta* study of
photosynthesis and photorespiration using NADPH and NADH/NAD⁺ fluorescent protein sensors. *Nat*
*Commun* 11 (3238), 3238.

Shimazaki, K.I., and Okayama, S. (1990). Calvin Benson cycle enzymes in guard-cell protoplasts and their
role in stomatal movement. *Biochem Physiol Pfl* 186, 327-331.

Voon, C.P., Guan, X., Sun, Y., Sahu, A., Chan, M.N., Gardeström, P., Wagner, S., Fuchs, P., Nietzel, T.,
Versaw, W.K., et al. (2018). ATP compartmentation in plastids and cytosol of *Arabidopsis thaliana*
revealed by fluorescent protein sensing. *Proc Natl Acad Sci U S A* 115, 10778-10787.

10.1073/pnas.1711497115.

Wang, S.W., Li, Y., Zhang, X.L., Yang, H.Q., Han, X.F., Liu, Z.H., Shang, Z.L., Asano, T., Yoshioka, Y., Zhang,
C.G., and Chen, Y.L. (2014). Lacking chloroplasts in guard cells of crumpled leaf attenuates stomatal
opening: both guard cell chloroplasts and mesophyll contribute to guard cell ATP levels. *Plant Cell and*
*Environment* 37, 2201-2210.

REVIEWER COMMENTS

Reviewer #1 (Remarks to the Author):

Lim, Flütsch and collaborators have substantially improved the work and performed additional experiments that provide important information for the regulation of both MCs and GCs metabolisms.

Congrats for this incredible work.

I only have minor comments, highlighted below.

General comment: The results of the Supplementary Fig 7d is very interesting, but not discussed. It seems that the import of ATP to GCCs through NTT1 is also important for dark-induced stomatal closure, not only light-induced stomatal movements. It would be great to have a brief explanation on how the lack of NTT1 (i.e. reduced GCCs ATP level) compromise the stomatal closure. This probably involves starch synthesis, given that both ntt1 and ntt2 have lower levels of starch at EoN. This idea is in agreement with the hypothesis raised by Outlaw in which gluconeogenesis and starch synthesis would be activated during stomatal closure.

Minor concern:

Lines 102-103: “We conclude that photosynthetic electron transport in Arabidopsis GCCs is limited”. I think the results of the Fig 1 do not support such conclusion. The higher stromal pH may be highly influenced by the higher levels of chlorophyll found in MCs. The level of ATP and NADPH is equal or even higher in GCs, compared to MCs at the same time points. It seems that the regulation of GC plastidial ATP and NADPH metabolisms is light independent, or not responsive to light exposition.

Minor comments:

Lines 144-145. “that mitochondria are a major source of cytosolic ATP of MCs and, particularly, GCs of mature leaves.” This is a very important result that contribute to solve a debated question in GCs. Furthermore, the results from MCs also contribute to explain previous modelling results from Lee Sweetlove group (Shameer et al., 2019).

Lines 212-214: “There are two possible carbon sources for GC starch synthesis: photoassimilates from GC photosynthesis and sugars imported from the mesophyll” Gluconeogenesis is likely another source of carbons for starch synthesis (Willmer and Dittrich, 1974; Outlaw and Kennedy, 1978; Lima et al., 2021)

Lines 380-381: “Fixation of CO₂ via PEP carboxylation ultimately leads to the formation of Mal²⁻, further highlighting the central role of this metabolite for GCs.” Indeed, this has been demonstrated through a recent positional ¹³C-analysis (Lima et al., 2021). The authors also detected ¹³C incorporation in glucose, including in the dark, suggesting that PEPc-mediated CO₂ assimilation is important for both sugar and organic acid homeostasis.

Lima VF, Erban A, Daubermann AG, et al. 2021. Establishment of a GC-MS-based ¹³C-positional isotopomer approach suitable for investigating metabolic fluxes in plant primary metabolism. *The Plant Journal*.

Outlaw WH, Kennedy J. 1978. Enzymic and substrate basis for the anaplerotic step in guard cells. *Plant physiology* 62, 648–652.

Shameer S, Ratcliffe RG, Sweetlove LJ. 2019. Leaf Energy Balance Requires Mitochondrial Respiration and Export of Chloroplast NADPH in the Light. *Plant Physiology* 180, 1947–1961.

Willmer CM, Dittrich P. 1974. Carbon dioxide fixation by epidermal and mesophyll tissues of *Tulipa* and *Commelina*. *Planta* 117, 123–132.

Reviewer #2 (Remarks to the Author):

The authors have made an excellent job addressing all of the reviewers’ comments and the MS has greatly improved. It is much clearer, the story more convincing and the conclusions drawn stronger.

Regarding the role of mitochondria in guard cell movement, the authors should include the recent paper by Violet-Charband et al., in *New Phytologist* that supports the role of mitochondria for energy release used in blue light induced stomatal opening.

The least convincing aspect of the MS for me remains the use of the probes and images provided in the MS to distinguish differences in ATP, NADPH and pH etc. It remains extremely difficult to see the differences highlighted in the bar charts in the actual images shown. I wonder if this is due to the large-scale bar range used. The majority of images only show the blue and green colour ranges, therefore if

the colour bar was re-scaled to these extremes, the contrast with the image would become more evident.

The authors demonstrate that 6-AN inhibits NADPH production in both gc and mc in the dark, however there would be no expectation of an alternative source in the dark. Does this not need to be carried out under illumination to explain the impact on stomatal opening?

Line 142 – the authors suggest more severe depletion of cytosolic ATP in the gc – however the control levels are already lower than MCs. Is the decrease statically greater between the two cell types?

Line 145 – It is not clear that these measurements were performed on leaves of different ages.

Suppl Fig.3 At 8h the images suggest that the GC are more alkaline after treatment with oligomycin than the controls.

There are several sections in the results where it would be useful for the authors to direct the reader back to the specific data. E.g. the first few sentences in the ntt1mutant section (starting line 182).

Can the authors provide an explanation as to why the ntt2 mutant has a high g_s despite lower starch, ATP etc. and the other features that provide the explanation for the ntt1 mutant function.

REVIEWER COMMENTS

Reviewer #1 (Remarks to the Author):

Lim, Flütsch and collaborators have substantially improved the work and performed additional experiments that provide important information for the regulation of both MCs and GCs metabolisms. Congrats for this incredible work.

I only have minor comments, highlighted below.

General comment: The results of the Supplementary Fig 7d is very interesting, but not discussed. It seems that the import of ATP to GCCs through NTT1 is also important for dark-induced stomatal closure, not only light-induced stomatal movements. It would be great to have a brief explanation on how the lack of NTT1 (i.e. reduced GCCs ATP level) compromise the stomatal closure. This probably involves starch synthesis, given that both *ntt1* and *ntt2* have lower levels of starch at EoN. This idea is in agreement with the hypothesis raised by Outlaw in which gluconeogenesis and starch synthesis would be activated during stomatal closure.

> This is a very good suggestion. Thank you. We have added the discussion about the importance of gluconeogenesis and ATP import to GCCs for starch synthesis and stomatal closure at lines 206-212.

Minor concern:

Lines 102-103: “We conclude that photosynthetic electron transport in Arabidopsis GCCs is limited”. I think the results of the Fig 1 do not support such conclusion. The higher stromal pH may be highly influenced by the higher levels of chlorophyll found in MCs. The level of ATP and NADPH is equal or even higher in GCs, compared to MCs at the same time points. It seems that the regulation of GC plastidial ATP and NADPH metabolisms is light independent, or not responsive to light exposition.
> Fig. 1 compared the dynamic changes of stromal ATP, NADPH and pH induced by illumination in GCs and MCs. We amended the sentence to “these data suggest that, comparing to MCCs, photosynthetic production of ATP and NADPH in Arabidopsis GCCs is limited.” (lines 99-100).

Minor comments:

Lines 144-145. “that mitochondria are a major source of cytosolic ATP of MCs and, particularly, GCs of mature leaves.” This is a very important result that contribute to solve a debated question in GCs. Furthermore, the results from MCs also contribute to explain previous modelling results from Lee Sweetlove group (Shameer et al., 2019).

> Thank you. We cited the paper (Shameer et al., 2019) in line 141 (Ref. 43).

Lines 212-214: “There are two possible carbon sources for GC starch synthesis: photoassimilates from GC photosynthesis and sugars imported from the mesophyll” Gluconeogenesis is likely another source of carbons for starch synthesis (Willmer and Dittrich, 1974; Outlaw and Kennedy, 1978; Lima et al., 2021)

> Thank you. We have included gluconeogenesis as a possible carbon source (lines 218-221) and cited the corresponding papers.

Lines 380-381: “Fixation of CO₂ via PEP carboxylation ultimately leads to the formation of Mal2-, further highlighting the central role of this metabolite for GCs.” Indeed, this has been demonstrated through a recent positional ¹³C-analysis (Lima et al., 2021). The authors also detected ¹³C incorporation in glucose, including in the dark, suggesting that PEPc-mediated CO₂ assimilation is important for both sugar and organic acid homeostasis.

> We thank the reviewer for this information. This study (Lima et al., 2021) is cited at lines 221, 383 and 387 (Ref. 52).

Lima VF, Erban A, Daubermann AG, et al. 2021. Establishment of a GC-MS-based ¹³C-positional isotopomer approach suitable for investigating metabolic fluxes in plant primary metabolism. *The Plant Journal*.

Outlaw WH, Kennedy J. 1978. Enzymic and substrate basis for the anaplerotic step in guard cells. *Plant physiology* 62, 648–652.

Shameer S, Ratcliffe RG, Sweetlove LJ. 2019. Leaf Energy Balance Requires Mitochondrial Respiration and Export of Chloroplast NADPH in the Light. *Plant Physiology* 180, 1947–1961.

Willmer CM, Dittrich P. 1974. Carbon dioxide fixation by epidermal and mesophyll tissues of *Tulipa* and *Commelina*. *Planta* 117, 123–132.

Reviewer #2 (Remarks to the Author):

The authors have made an excellent job addressing all of the reviewers' comments and the MS has greatly improved. It is much clearer, the story more convincing and the conclusions drawn stronger. Regarding the role of mitochondria in guard cell movement, the authors should include the recent paper by Violet-Charband et al., in *New Phytologist* that supports the role of mitochondria for energy release used in blue light induced stomatal opening.

> Thank you. We have cited the work in line 287 (Ref. 54).

The least convincing aspect of the MS for me remains the use of the probes and images provided in the MS to distinguish differences in ATP, NADPH and pH etc. It remains extremely difficult to see the differences highlighted in the bar charts in the actual images shown. I wonder if this is due to the large-scale bar range used. The majority of images only show the blue and green colour ranges, therefore if the colour bar was re-scaled to these extremes, the contrast with the image would become more evident.

> We thank reviewer for pointing out his doubt. Although these genetically encoded fluorescent proteins are relatively new in plant field, all sensors used in this manuscript have been fully characterized in *Arabidopsis* and have been used by several research groups to answer scientific questions (e.g iNAPs (Lim et al., 2020; Haber et al., 2021), cpYFP (Behera et al., 2018; Voon et al., 2018; Lim et al., 2020), AT1.03 (De Col et al., 2017; Voon et al., 2018)).

To avoid errors during image processing, the parameters for processing the ratio images given in this manuscript (for example, Supplementary Figure 1, Supplementary Figure 3) are the same as the settings of our previous publications. We included a note to the figure legends to inform the reader that some of the images are raw ratio but not normalized ratios. As a result, the ratios are not explicitly implied by the displayed images.

All the raw image data are now available to the public through the repository BioStudies (line 570).

<https://www.ebi.ac.uk/biostudies/>

The accession number is TMP_1638269162656, which will be online on 1.1.2022.

The authors demonstrate that 6-AN inhibits NADPH production in both gc and mc in the dark, however there would be no expectation of an alternative source in the dark. Does this not need to be carried out under illumination to explain the impact on stomatal opening?

> We thank the reviewer for this important comment. We actually used 6-aminonicotinamide (6-AN) to illustrate that OPPP is indeed the major source of NADPH in MC and GC in the dark. It is well-known that the OPPP pathway is inactive in the light, as the first enzyme of OPPP, glucose-6-

phosphate dehydrogenase (G6PDH), which generates NADPH, is inactivated by light-activated thioredoxins (Née et al., 2009).

Line 142 – the authors suggest more severe depletion of cytosolic ATP in the gc – however the control levels are already lower than MCs. Is the decrease statically greater between the two cell types?

> We apologise if the wording created confusion. As we did not directly compare the two cell types, we removed the sentence “but the effect was more severe in GCs, which were almost depleted of cytosolic ATP” (Line 139).

Line 145 – It is not clear that these measurements were performed on leaves of different ages. Suppl Fig.3 At 8h the images suggest that the GC are more alkaline after treatment with oligomycin than the controls.

>The measurements were carried out using the 4th leaves of 20- to 22-day-old plants. It is now stated in the figure legend of Supplementary Fig. 3.

Yes, at 8h the GC cytosol is more alkaline after oligomycin treatment. One possible explanation is that since oligomycin inhibits ATP production from mitochondria, the GC at 8h needs to increase ATP production via glycolysis (of its accumulated starch or imported sugars). However, since pyruvate is not consumed by mitochondria anymore, accumulated pyruvate has to be converted into acetaldehyde and then to ethanol to regenerate NAD⁺. These enzymatic processes consume protons and will alkalize the cytosol.

As cytosolic pH can be affected by many biological processes and it is not the main topic of this manuscript, we do not want to over-interpret this observation. Nonetheless, our data showed that alkalization of cytosolic pH induced by illumination is abolished in both MCs and GCs after oligomycin treatment.

There are several sections in the results where it would be useful for the authors to direct the reader back to the specific data. E.g. the first few sentences in the *ntt1* mutant section (starting line 182).

> We apologise for the inconvenience. Figure numbers were added to the sentences in lines 163, 180, 198, 203 and 204.

Can the authors provide an explanation as to why the *ntt2* mutant has a high *g_s* (Fig. 5d) despite lower starch, ATP etc. and the other features that provide the explanation for the *ntt1* mutant function.

> Based on qPCR data, it is clear that NTT1 is the major isoform of NTTs in GCs. This is reflected in the phenotypes of the corresponding *ntt* mutants. Despite the trend of *g_s* in *ntt2* is higher than that of WT, there are no statistically significant differences between the mutant and the WT, as highlighted in the figure itself. While it is true that *ntt2* has lower starch content in GCs compared to WT, we still see a certain degree of starch turnover, and most importantly a statically significant decrease in GC starch content upon illumination, which may explain the *g_s* trend. It is possible that *NTT1* is upregulated in *ntt2* mutant background to compensate for its absence, although we have not tested it directly. We briefly discuss this point in the corresponding result section.

Behera, S., Xu, Z.L., Luoni, L., Bonza, M.C., Doccua, F.G., De Michelis, M.I., Morris, R.J., Schwarzlander, M., and Costa, A. (2018). Cellular Ca²⁺ Signals Generate Defined pH Signatures in Plants. *Plant Cell* **30, 2704-2719.**

- De Col, V., Fuchs, P., Nietzel, T., Elsässer, M., Voon, C.P., Candeo, A., Seeliger, I., Fricker, M.D., Grefen, C., Møller, I.M., Bassi, A., Lim, B.L., Zancani, M., Meyer, A.J., Costa, A., Wagner, R., and Schwarzländer, M.** (2017). ATP sensing in living plant cells reveals tissue gradients and stress dynamics of energy physiology. *eLife* **6**, 26770.
- Haber, Z., Lampl, N., Meyer, A.J., Zelinger, E., Hipsch, M., and Rosenwasser, S.** (2021). Resolving diurnal dynamics of the chloroplastic glutathione redox state in *Arabidopsis* reveals its photosynthetically-derived oxidation. *The Plant Cell* **33**, 1828-1844.
- Lim, S.L., Voon, C.P., Guan, X., Yang, Y., Gardestrom, P., and Lim, B.L.** (2020). *In planta* study of photosynthesis and photorespiration using NADPH and NADH/NAD⁺ fluorescent protein sensors. *Nat Commun* **11**, 3238.
- Lima, V.F., Erban, A., Daubermann, A.G., Freire, F.B.S., Porto, N.P., Cândido-Sobrinho, S.A., Medeiros, D.B., Schwarzländer, M., Fernie, A.R., dos Anjos, L., Kopka, J., and Daloso, D.M.** (2021). Establishment of a GC-MS-based ¹³C-positional isotopomer approach suitable for investigating metabolic fluxes in plant primary metabolism. *The Plant Journal* **108**, 1213-1233.
- Née, G., Zaffagnini, M., Trost, P., and Issakidis-Bourguet, E.** (2009). Redox regulation of chloroplastic glucose-6-phosphate dehydrogenase: A new role for f-type thioredoxin. *FEBS letters* **583**, 2827-2832.
- Shameer, S., Ratcliffe, R.G., and Sweetlove, L.J.** (2019). Leaf energy balance requires mitochondrial respiration and export of chloroplast NADPH in the light. *Plant Physiology* **180**, 1947-1961.
- Voon, C.P., Guan, X., Sun, Y., Sahu, A., Chan, M.N., Gardeström, P., Wagner, S., Fuchs, P., Nietzel, T., Versaw, W.K., Schwarzländer, M., and Lim, B.L.** (2018). ATP compartmentation in plastids and cytosol of *Arabidopsis thaliana* revealed by fluorescent protein sensing. *Proc Natl Acad Sci U S A* **115**, 10778-10787.

REVIEWERS' COMMENTS

Reviewer #1 (Remarks to the Author):

The authors have covered all my concerns. I believe the work is now suitable for publication in Nat Comm.

Reviewer #2 (Remarks to the Author):

The authors have made an excellent job of addressing all of the reviewers comments. I am satisfied with the corrections.